# Transcriptional rewiring over evolutionary timescales changes quantitative and qualitative properties of gene expression

**Chiraj K Dalal[1]\*, Ignacio A Zuleta[2,3], Kaitlin F Mitchell[4,5], David R Andes[4,5], Hana El-Samad[2,3], Alexander D Johnson[1,6]\***

[1]Department of Microbiology and Immunology, University of California, San Francisco, San Francisco, United States; [2]Department of Biochemistry and Biophysics, University of California, San Francisco, San Francisco, United States; [3]California Institute for Quantitative Biosciences, University of California, San Francisco, San Francisco, United States; [4]Department of Medicine, University of Wisconsin, Madison, United States; [5]Department of Medical Microbiology and Immunology, University of Wisconsin, Madison, United States; [6]Department of Biochemistry and Biophysics, University of California, San Francisco, San Francisco, United States

**Abstract** Evolutionary changes in transcription networks are an important source of diversity across species, yet the quantitative consequences of network evolution have rarely been studied. Here we consider the transcriptional 'rewiring' of the three *GAL* genes that encode the enzymes needed for cells to convert galactose to glucose. In *Saccharomyces cerevisiae*, the transcriptional regulator Gal4 binds and activates these genes. In the human pathogen *Candida albicans* (which last shared a common ancestor with *S. cerevisiae* some 300 million years ago), we show that different regulators, Rtg1 and Rtg3, activate the three *GAL* genes. Using single-cell dynamics and RNA-sequencing, we demonstrate that although the overall logic of regulation is the same in both species—the *GAL* genes are induced by galactose—there are major differences in both the quantitative response of these genes to galactose and in the position of these genes in the overall transcription network structure of the two species.

**\*For correspondence:** chirajdalal.
ucsf@gmail.com (CKD); ajohnson@
cgl.ucsf.edu (ADJ)

**Competing interests:** The authors declare that no competing interests exist.

## Introduction

Gene regulatory networks undergo significant divergence over evolutionary time (*Carroll, 2005*; *Davidson, 2006*; *Doebley and Lukens, 1998*; *Tuch et al., 2008*; *Wohlbach et al., 2009*; *Wray, 2007*). Except for the simplest cases of changes in the regulation of a single gene (*Chan et al., 2010*; *Gompel et al., 2005*; *Tishkoff et al., 2007*), we do not fully understand how these evolutionary changes occur or how they impact modern species. In particular, little attention has been paid to the quantitative consequences of transcriptional rewiring. Here we describe an evolutionary analysis of a transcriptional circuit controlling the production of enzymes that convert galactose to glucose-1-phosphate via the Leloir pathway (*Kew and Douglas, 1976*); these enzymes (a kinase, epimerase and transferase) are conserved across all kingdoms of life (*Holden et al., 2003*).

In most organisms, expression of the three Leloir enzymes increases when galactose is present in the growth medium (*Holden et al., 2004*); this regulation has been extensively studied in the budding yeast *S. cerevisiae* (*Campbell et al., 2008*; *Conrad et al., 2014*; *Giniger et al., 1985*; *Johnston, 1987*; *Lohr et al., 1995*; *Ptashne and Gann, 2003*; *Sellick and Reece, 2006*; *Traven et al.,*

*2006*). Here the zinc cluster transcriptional regulator Gal4 binds to short sequence motifs in the upstream region of the genes encoding these enzymes (*GAL1, GAL7* and *GAL10*) and induces their transcription when galactose is present and glucose is absent. The expression of *GAL1* has been estimated to be <1 mRNA molecules/10 cells in glucose and ~35 mRNA molecules/cell in galactose (*Iyer and Struhl, 1996*). Due in part to this high induction ratio (>350 fold), Gal4 has been adapted as a tool to artificially turn genes on and off in many animal and plant species (*Brand and Perrimon, 1993*; *Fischer et al., 1988*; *Hartley et al., 2002*; *Kakidani and Ptashne, 1988*; *Waki et al., 2013*; *Webster et al., 1988*).

The *Candida albicans* genome contains an unmistakable ortholog of each Leloir pathway enzyme (*Brown et al., 2009*; *Fitzpatrick et al., 2010*; *Martchenko et al., 2007a*), arranged in a cluster as they are in *S. cerevisiae* (*Fitzpatrick et al., 2010*; *Slot and Rokas, 2010*). It also contains a clear ortholog of the transcriptional regulator Gal4 (*Martchenko et al., 2007b*; *Sellam et al., 2010*). However, in *C. albicans*, which last shared a common ancestor with *S. cerevisiae* at least 300 million years ago (*Taylor and Berbee, 2006*), Gal4 does not play a role in expressing the three *GAL* enzymes (*Martchenko et al., 2007a*); instead, it has been implicated as having a subsidiary role in regulating glucose utilization (*Askew et al., 2009*). Despite being uncoupled from Gal4, the three *GAL* genes in *C. albicans* are transcriptionally activated when galactose is present and glucose is absent in the growth medium (*Brown et al., 2009*). While a few fungal species have lost the *GAL1, GAL7*, and *GAL10* gene cluster entirely, most have retained it, including several pathogenic species closely related to *C. albicans* such as *C. dubliniensis, C. tropicalis* and *C. parapsilosis* (*Fitzpatrick et al., 2010*; *Hittinger et al., 2004*; *Slot and Rokas, 2010*). Since the only known environmental niche for *C. albicans* is in or on warm-blooded animals (*Odds, 1988*), it seems very likely that the three *GAL* genes and their regulation is important for the ability of *C. albicans* to survive in its host.

In this paper, we use a variety of experimental and bioinformatics approaches to establish the mechanisms through which the *GAL1, GAL7*, and *GAL10* genes are transcriptionally activated in *C. albicans*. By considering outgroup species, we have also inferred the order of several key events that led to the difference in the circuitry between *S. cerevisiae* and *C. albicans*. Finally, we compare the quantitative output of the different regulatory schemes used in *C. albicans* and *S. cerevisiae* and document several striking differences.

## Results

### Identification of galactose metabolism circuit components in *C. albicans*

We experimentally verified that the closest matches to the *GAL1, GAL7* and *GAL10* genes in *C. albicans* did indeed code for the enzymes necessary for galactose metabolism. Each of these genes was deleted individually and the resulting mutants were tested for growth on media that included galactose as the sole sugar (*C. albicans* is diploid, so two rounds of disruption were needed per gene [*Hernday et al., 2010*]). To force the cells to ferment galactose in order to grow, we included the respiration inhibitor Antimycin A (*Askew et al., 2009*). We found that the *C. albicans* parent strain grows normally under these conditions but none of the three knockout strains could grow in the presence of Antimycin A and galactose as the sole sugar (*Figure 1B*, *Figure 1—figure supplement 1*), a behavior similar to *S. cerevisiae GAL* mutants (*Dudley et al., 2005*). Growth in glucose was unaffected by the deletions. From these results, we conclude that the *C. albicans GAL1, GAL7* and *GAL10* genes are indeed the functional orthologs of the *S. cerevisiae GAL* genes.

We next considered whether *GAL1, GAL7* and *GAL10* are required for *C. albicans* to proliferate in different animal models of infection. Previous experiments in mice have shown that Gal10, but not Gal1, is required for *C. albicans* to proliferate in a commensal (gut colonization) model of infection, while neither protein is required for *C. albicans* to disseminate in a systemic (tail-vein injection) model of infection (*Pérez et al., 2013*). Here we tested whether the *GAL1, GAL7*, and *GAL10* genes are required for colonization in a rat catheter model. In this infection model, a catheter is placed in the jugular vein of a rat and *C. albicans* strains are introduced to measure their ability to colonize the catheter by forming a biofilm (*Nett et al., 2012*); this model was designed to recapitulate catheter infections in humans and has been extensively validated (*Nett and Andes, 2015*; *Nobile et al., 2012*). We found that the knockout strains of *GAL1, GAL7* and *GAL10* all showed severe defects (compared to a matched parent strain) in this infection model (*Figure 2*, *Figure 2—figure*

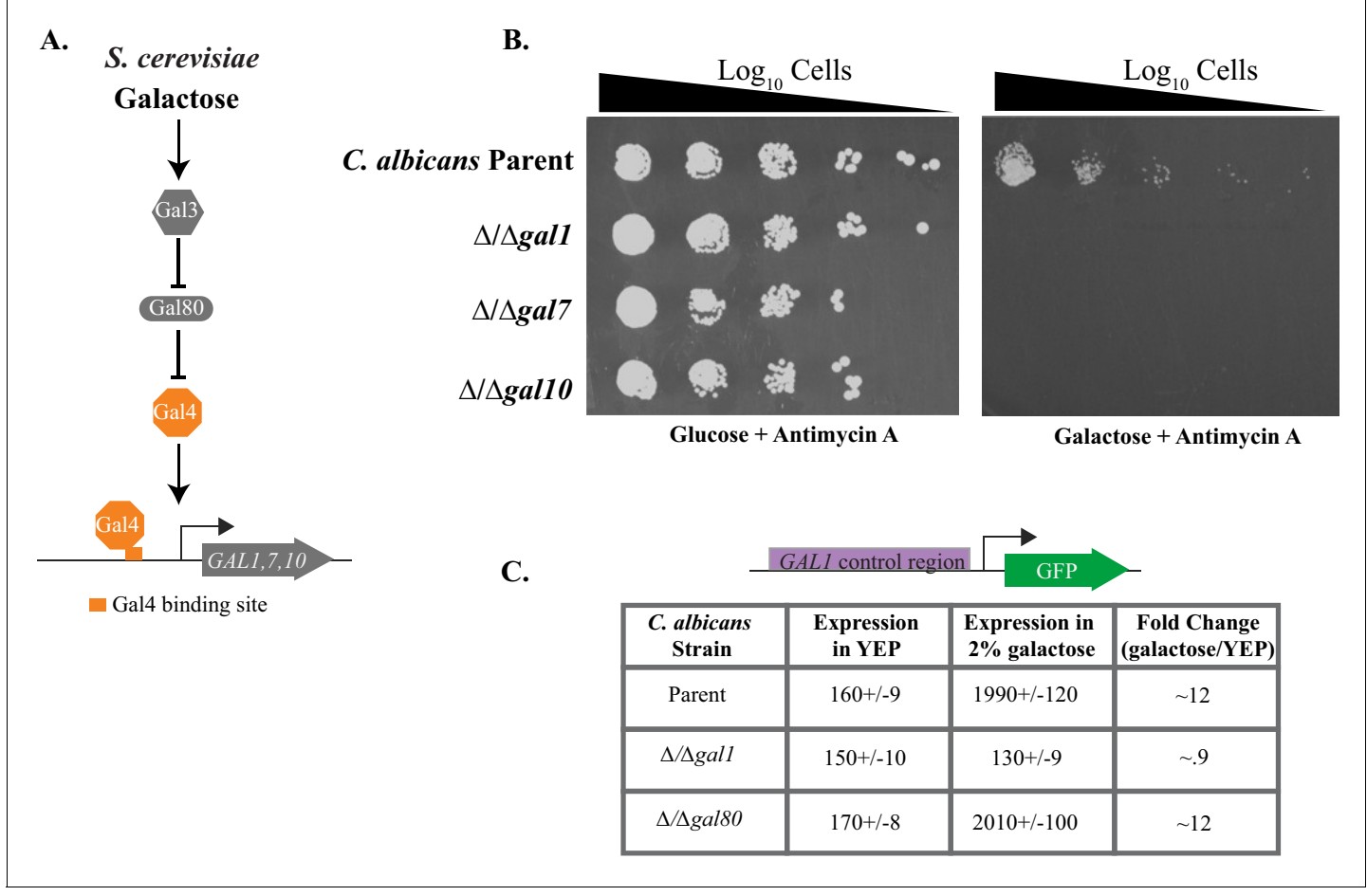

**Figure 1.** *C. albicans GAL* genes are needed for efficient growth on galactose. (**A**) A schematic of *GAL* gene regulation in *S. cerevisiae*. (**B**) $Log_{10}$ serial dilutions of a *C. albicans* parent strain (top) and isogenic strains deleted for *GAL1, GAL7,* and *GAL10* were spotted onto plates containing 2% glucose + 3 µg/ml Antimycin A (left panel) and 2% galactose + 3 µg/ml Antimycin A (right panel). Images were acquired 6 days after growth at 30℃. The same behavior was observed for independently constructed knockout strains, as shown in *Figure 1—figure supplement 1*. (**C**) A GFP reporter strain was constructed by precisely substituting one copy of the *GAL1* ORF with the *GFP* ORF. GFP expression of this reporter was monitored in a *C. albicans* parent strain, and compared with expression from isogenic strains deleted for *GAL1* and *orf19.6899* (a potential ortholog of *S. cerevisiae GAL80*). Expression was measured by flow cytometry after 6 hr of growth in the indicated media. Mean expression levels are reported for each measurement in arbitrary units. Errors indicate standard errors derived from three independent measurements.

The following figure supplement is available for figure 1:

**Figure supplement 1.** Antimycin A spotting assays in independently constructed strains *of C. albicans*.

*supplement 1*). These were some of the most pronounced defects observed for any previously studied gene knockout in this *C. albicans* biofilm model, and the results clearly show that the *GAL1, GAL7,* and *GAL10* genes are required for this well-characterized colonization model.

We next turn to the regulators of the *GAL* enzymes. As a result of a whole genome 'duplication,' now known to be a hybridization between two closely related species (*Marcet-Houben and Gabaldón, 2015*), *S. cerevisiae* has a paralog of *GAL1*, called *GAL3*, which plays a signaling role in activating the *GAL* genes in *S. cerevisiae* (*Bhat and Murthy, 2001*; *Hittinger and Carroll, 2007*; *Sellick and Reece, 2006*, *Figure 1A*). However, *C. albicans* branched before this duplication and has only a single *GAL1*. To test whether the *C. albicans* Gal1 serves as both an enzyme (like Gal1) and an upstream signaling component (like Gal3), we measured the expression of the *GAL1* promoter using a GFP reporter in a strain deleted for *GAL1*. In this reporter (p*GAL1*-GFP), we replaced the *GAL1* open reading frame with GFP (*Cormack et al., 1997*). Expression of the reporter was

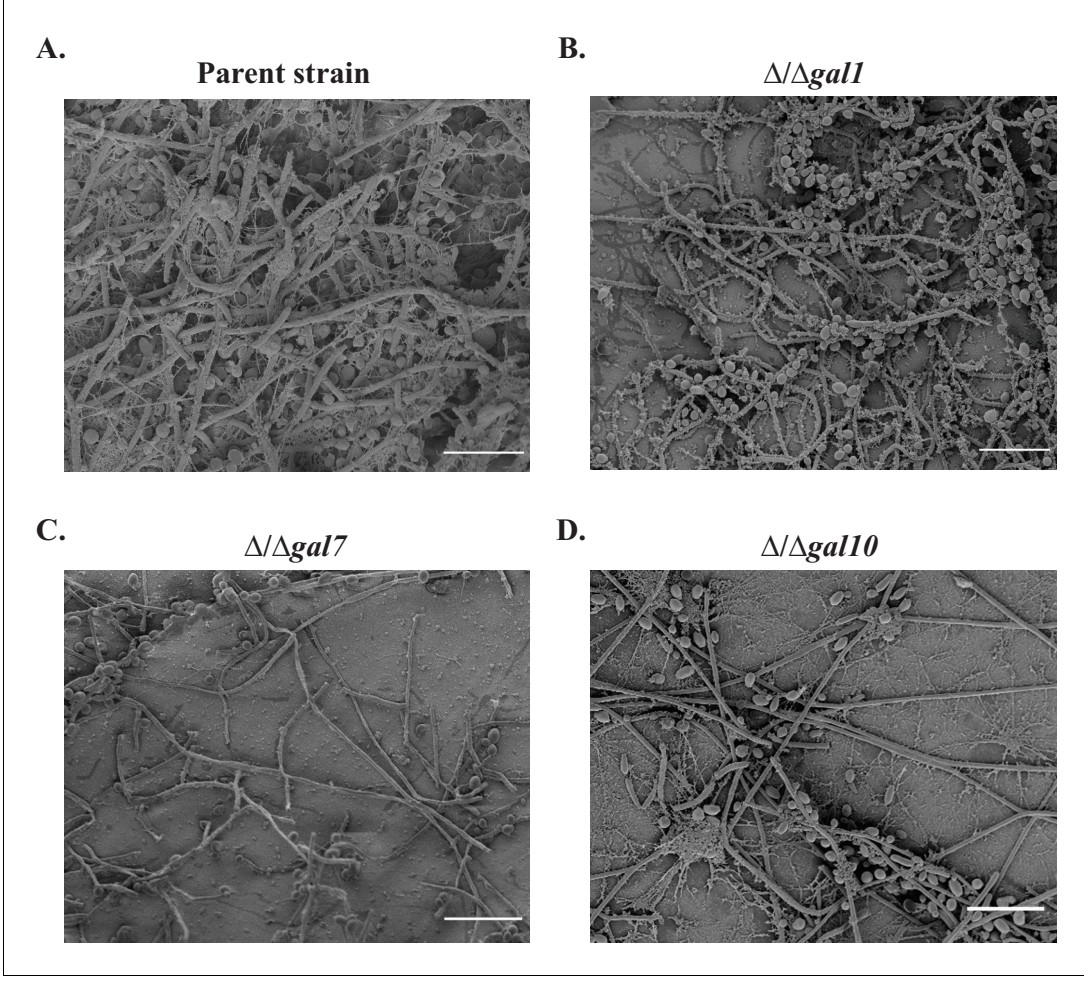

**Figure 2.** The *GAL* genes are required for colonization in an *in vivo* rat catheter model of infection. (**A–D**) The parent strain (**A**) and the strains in which *GAL1, GAL7,* and *GAL10* were deleted (**B–D**) were inoculated into rat intravenous catheters; resulting biofilms were visualized after 24 hr of growth by scanning electron microscopy. The images show the catheter luminal surfaces at 1000x magnification. The scale bar represents 20 µm.

The following figure supplement is available for figure 2:

**Figure supplement 1.** Biofilm formation in the *in vivo* rat catheter model.

significantly lower in the presence of galactose (*Figure 1C*), indicating that Gal1 has an activating upstream signaling role in *C. albicans*, in addition to its enzymatic role. This situation is similar to that in *K. lactis*, another 'pre whole-genome hybridization' species (*Anders and Breunig, 2011*; *Hittinger and Carroll, 2007*; *Meyer et al., 1991*; *Rubio-Texeira, 2005*).

The *C. albicans* transcriptional regulator Gal4 is clearly orthologous to *S. cerevisiae* Gal4, a conclusion supported by extensive phylogenetic analysis (*Martchenko et al., 2007b*) as well as by the fact that the Gal4 protein from both species has the same DNA-binding specificity (*Askew et al., 2009*). As discussed above, Gal4 is not required for regulation of the *GAL* enzymes in *C. albicans* (*Martchenko et al., 2007a*). We note that *C. albicans* also contains a gene (*orf19.6899*) with ~40% amino acid identity to *S. cerevisiae* Gal80 (*Wapinski et al., 2007*), the regulatory protein that prevents Gal4 from activating the *GAL* genes in the absence of galactose (*Torchia et al., 1984*) (*Figure 1A*). To test whether this gene has a role in galactose regulation in *C. albicans*, we tagged one copy of the *GAL1* promoter with GFP (*Cormack et al., 1997*) and measured gene expression in parent and Δ/Δ *orf19.6899* mutant cells. We found that the expression of *GAL1* in parent and knockout cells was similar in response to galactose and glucose (*Figure 1C*), demonstrating that, like Gal4,

the Gal80 ortholog does not play a significant role in regulating the *GAL1, GAL7*, and *GAL10* genes in *C. albicans*.

Taken together, the experiments described above, in combination with observations from the literature, establish that (1) *C. albicans GAL1, GAL7* and *GAL10* orthologs encode enzymes needed to convert galactose to glucose-1 phosphate, (2) neither Gal4 nor Gal80, the key *GAL* regulators in *S. cerevisiae*, control the *GAL* genes in *C. albicans* and (3) like the pre-hybridization species *K. lactis*, the *C. albicans GAL1* gene doubles as both an enzyme and a signaling component.

## Identification of transcriptional regulator(s) controlling galactose metabolism in *C. albicans*

Given that Gal4 is not needed to activate the three *C. albicans GAL* genes in response to galactose (*Askew et al., 2009*; *Martchenko et al., 2007a*), some other transcriptional regulator(s) must carry out this function. To identify this protein, we utilized a collection of transcription factor knockout strains in *C. albicans* (*Fox et al., 2015*; *Homann et al., 2009*) and assayed them for growth defects on media containing Antimycin A (to prevent respiration) and galactose as the only sugar. We screened 212 knockout strains and found only two, △/△rtg1 and △/△rtg3, that grew several orders of magnitude slower than the parent strain under these conditions (*Figure 3A–B, Figure 1—figure supplement 1*). This response is specific to galactose as these cells grow at normal levels on glucose, even in the presence of Antimycin A (*Figure 3A*). Adding back the *RTG1* and *RTG3* genes to their respective knockout strains restores growth on galactose + Antimycin A to levels comparable to the parent strain (*Figure 3A, Figure 1—figure supplement 1*). We note that strains knocked out for *CPH1* and *RGT1*, transcriptional regulators previously implicated in galactose metabolism in *C. albicans* (*Brown et al., 2009*; *Martchenko et al., 2007a*), were represented in our library but did not display a significant galactose-specific growth defect (*Figure 3A, Figure 3B—source data 1*).

To test whether Rtg1 and Rtg3 regulate expression of the three *GAL* genes in *C. albicans*, we measured the *GAL1*-GFP expression (using a strain where the GFP coding region precisely replaced the *GAL1* coding region in one of the two *GAL1* copies) in each of the single △/△rtg1 and △/△rtg3 knockout strains as well as in a △/△rtg1 △/△rtg3 double mutant we constructed. *GAL1* expression in the presence of galactose was significantly reduced in the △/△rtg1 strain and in the double mutant (*Figure 3C*). Although the *RTG3* deletion strain had a profound growth defect on galactose (*Figure 3A, Figure 1—figure supplement 1B*), the deletion showed no significant effect on the steady-state levels of *GAL1* expression (*Figure 3C*).

In *S. cerevisiae*, Rtg1 and Rtg3 are basic helix-loop-helix transcriptional regulators that form a heterodimer to regulate metabolic signaling in response to mitochondrial dysfunction (*Jia et al., 1997*). The two proteins are similar to each other in amino acid sequence, with 34% identity. Knockouts of *RTG1* in *S. cerevisiae* strongly reduce this regulation while the *RTG3* knockouts show a milder effect (*Hashim et al., 2014*; *Kemmeren et al., 2014*). We note that the *rtg1* and *rtg3* knockout strains were independently identified 'blindly' in our *C. albicans* screen, providing further evidence that the two proteins work together as they do in *S. cerevisiae*.

## Do Rtg1 and Rtg3 directly regulate the *GAL* genes in *C. albicans*?

We scanned the upstream regions (up to 800 base pairs) of the *GAL1, GAL7* and *GAL10* genes in *C. albicans* and the orthologous *GAL* genes from five closely related members of the CTG clade (*Figure 3—figure supplement 1*) to search for conserved *cis*-regulatory elements (*Carlson et al., 2007*; *Martyanov and Gross, 2011*). The CTG clade, so named because the CTG codon is translated as serine instead of a conventional leucine, represents approximately 170 million years of divergence (*McManus and Coleman, 2014*). We compared these results to similar (in essence, control) scans in *S. cerevisiae* and five other members of the *Saccharomycotina* and *Kluyveromyces* clades. As expected, the top-scoring motif from the *GAL* genes in the *Saccharomycotina* and *Kluyvermoyces* clades was the well-documented Gal4 motif (*Guarente et al., 1982*), with two-half sites spaced 11 nucleotides apart (*Figure 3—figure supplement 1*). This motif was not detected in the three *GAL* genes of *C. albicans* or any member of the CTG clade. Instead, the top motif in these 6 species is a longer palindromic motif, similar to the Rtg1-Rtg3 motif that was identified in *C. albicans* (*Pérez et al., 2013*) and *S. cerevisiae* (*Jia et al., 1997, Figure 3—figure supplement 1*).

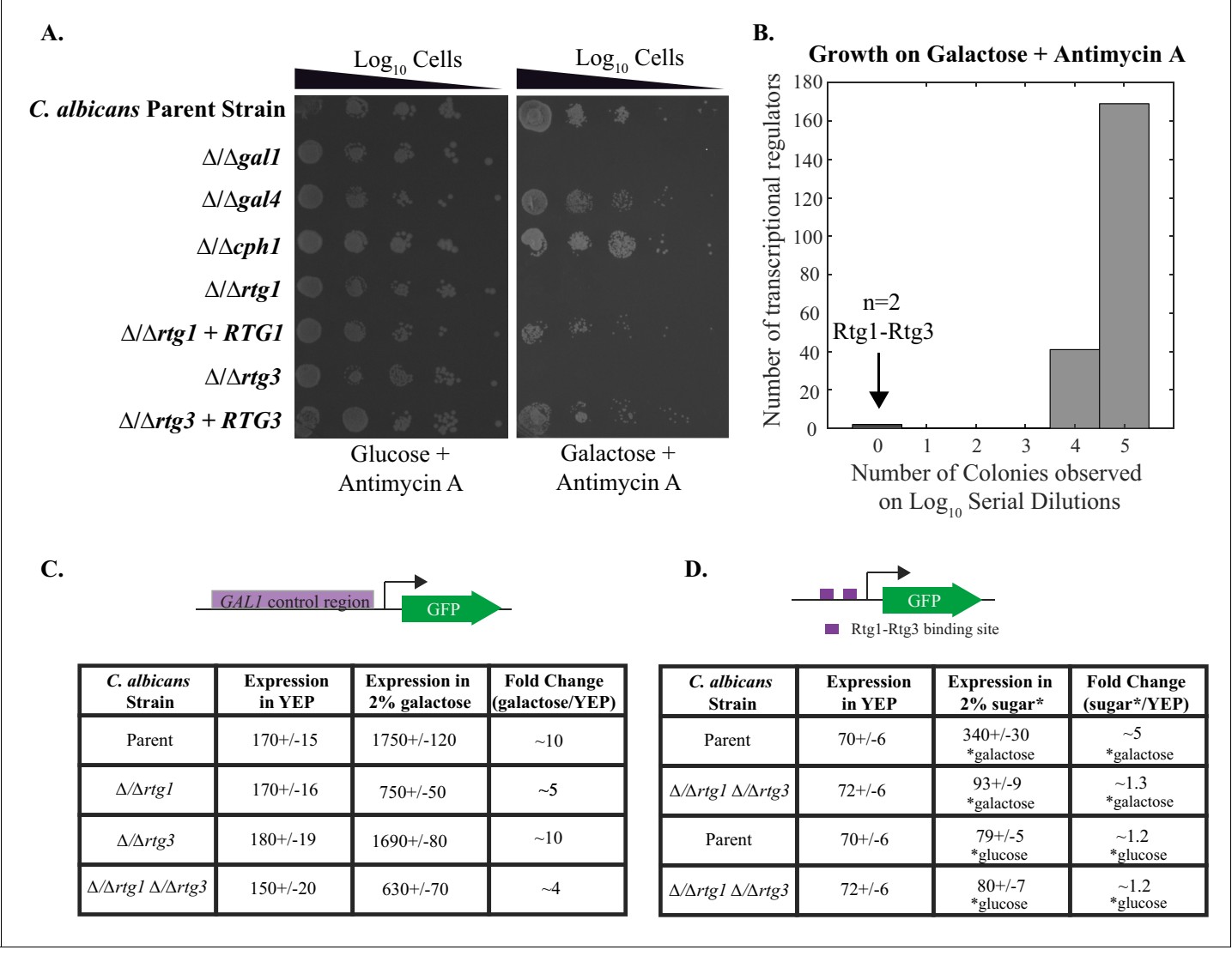

**Figure 3.** Rtg1-Rtg3 regulate galactose-mediated activation of the *GAL* genes in *C. albicans*. (A) Log$_{10}$ serial dilutions of a *C. albicans* parent strain (top) and isogenic strains deleted for *GAL1*, *GAL4*, *CPH1*, *RTG1* and *RTG3* were spotted onto plates containing 2% glucose + 3 µg/ml Antimycin A (left panel) and 2% galactose + 3 µg/ml Antimycin A (right panel). Also included are several gene 'addback' strains where the indicated gene was reintroduced into the corresponding deletion strain. Images were acquired 6 days after growth at 30°C. See ***Figure 1—figure supplement 1*** for images of independently constructed isolates. (B). Log$_{10}$ serial dilutions of 212 transcription factor knockout strains were spotted onto plates containing 2% galactose + 3 µg/ml Antimycin A. The numbers of dilutions where colonies were observed were tabulated and plotted as a histogram. The results show that, of the 212 deletion strains, only Δ/Δ*rtg1* and Δ/Δ*rtg3* had severe growth defects under this condition (See ***Figure 3—source data 1*** for complete data). Neither strain showed a growth defect on plates containing 2% glucose + 3 µg/ml Antimycin A. (C) GFP expression driven by the *GAL1* upstream region in a *C. albicans* parent strain and isogenic strains deleted for *RTG1*, *RTG3* or both were measured by flow cytometry after 6 hr of growth in media containing 2% galactose. Mean expression levels are reported in arbitrary units with standard errors derived from three independent measurements. (D) Rtg1-Rtg3 consensus *cis*- regulatory motifs (found in all three of the *C. albicans GAL1*, *GAL7* and *GAL10* regulatory regions) were synthesized and ligated into a promoter lacking upstream regulatory sequences, coupled to a GFP reporter. This construct was integrated into the parent strain and isogenic strains deleted for both *RTG1* and *RTG3*. GFP fluorescence was measured by flow cytometry after 6 hr of growth in YEP media with and without the indicated sugar. Mean expression levels are reported for each sugar in arbitrary units. Errors indicate standard errors from three independent measurements.

The following source data and figure supplement are available for figure 3:

**Source data 1.** Antimycin A spotting results from TFKO screen.
**Figure supplement 1.** Motif analysis of *GAL* genes.

Two experiments confirm that this Rtg1-Rtg3 motif is responsible for regulating the three *GAL* genes in *C. albicans*. First, previous whole-genome ChIP experiments from our lab (carried out for entirely different purposes) showed Rtg1 and Rtg3 binding peaks over the Rtg1-Rtg3 motifs in the upstream regions of all three *GAL* genes (*Pérez et al., 2013*, *Figure 3—figure supplement 1*). Second, we tested whether the Rtg1-Rtg3 motif was sufficient to bring about galactose-induced transcription in *C. albicans*. We inserted two copies of a consensus Rtg1-Rtg3 motif (derived from the upstream regions of the three *C. albicans GAL* genes, *Figure 3—figure supplement 1*) into a truncated *CYC1* promoter driving GFP and measured gene expression in cells grown in glucose and galactose. The construct was induced by galactose, compared to no sugar (*Figure 3D*); this activation was absent when the motif was omitted from the construct or when *RTG1* and *RTG3* were deleted from the genome (*Figure 3D*). We note that the Rtg1-Rtg3 motifs, although sufficient for induction by galactose, do not completely recapitulate the behavior of the natural p*GAL1* promoter; in particular, the Rtg1-Rtg3 motif construct was not repressed in the presence of glucose. This observation indicates that additional *cis*-regulatory sequences present in the natural *GAL1* promoter are needed for glucose repression. Taken together, the results of the genetic screen, the similarity of the *C. albicans GAL* gene *cis* regulatory element to the *S. cerevisiae* Rtg1-Rtg3 binding site, the ChIP experiment in *C. albicans,* and the reporter expression experiments show that Rtg1 and Rtg3 are responsible for the galactose-inducible expression of the three *C. albicans GAL* genes by binding directly to the *cis*-regulatory sequences located upstream of the genes.

As mentioned above, we do not believe that *RTG1* and *RTG3* are the only regulators of the *GAL* genes in *C. albicans*. Their binding motif alone produces a 5-fold induction of galactose which is almost completely dependent on Rtg1-Rtg3; however glucose repression of the *GAL* genes is not recapitulated from this motif and likely lies at control sequences outside of it. In addition, the intact *GAL1* regulatory region exhibits a 12-fold induction, some of which still remains in the △/△*rtg1* △/△*rtg3* double mutant. For example, it is possible that Rgt1 [a transcriptional regulator implicated in the study of *Brown et al. (2009)*] and/or Cph1 [implicated by *Martchenko et al. (2007a)*] contributes to this residual induction, even though neither deletion strain shows a growth defect on galactose (*Figure 3—source data 1*).

## Measuring circuit output of *GAL* gene expression in *S. cerevisiae* and *C. albicans*

To determine whether the transcriptional rewiring of the *GAL* genes between *S. cerevisiae* and *C. albicans* has an impact on the quantitative output of the circuit in each species, we compared the expression dynamics of *GAL1* between the W303 strain of *S. cerevisiae* and the SC5314 strain of *C. albicans*. In each species, the *GAL1* promoter was fused to GFP; to optically distinguish between the species, *C. albicans* was engineered to also express a constitutive fluorescent protein, Rpl26b-mCherry (*Figure 4A*). These two species were grown individually in media lacking a sugar source; they were then combined in equal proportions in 96 different wells, each containing a different combination of galactose and glucose (*Figure 4D*). Expression of the reporters in single cells was continuously monitored in these populations every 20 min for ~10 hr using an automated flow cytometry-based fermentation system that allows continuous sampling (*Zuleta et al., 2014*).

These experiments confirm that, despite the evolutionary wiring change, the basic logic of the galactose circuit is preserved between *C. albicans* and *S. cerevisiae*: *GAL1* expression increases in galactose and decreases in glucose (*Figure 4C*). However, there are several obvious differences in the quantitative behaviors of the two circuits. (1) The dynamic range of activation (ON vs OFF) is much larger in *S. cerevisiae* (~900 fold) than in *C. albicans* (~12 fold, *Figure 4B–D*, *Figure 4—figure supplement 1A*; *Figure 4—figure supplement 1B*). Two factors contribute to this difference: the basal expression of *GAL1* (expression in the presence of glucose and absence of galactose) is higher in *C. albicans*, and the maximum expression level in *C. albicans* (in the presence of galactose and absence of glucose) is lower than in *S. cerevisiae*. (2) In the absence of glucose, *GAL1* begins its induction 20–40 min faster in *C. albicans* than in *S. cerevisiae* (*Figure 4*, *Figure 4—figure supplement 1*). These conclusions are based the first derivative of fluorescence (*Figure 4—figure supplement 1*) and rely on the assumption that the rates of GFP folding in *S. cerevisiae* and *C. albicans* do not vary sufficiently to account for these differences. (3) The third major difference in the *C. albicans* and *S. cerevisiae* response is the concentration of galactose needed for half-maximal activation of *GAL1*; it is ~15 fold lower in *C. albicans* than in *S. cerevisiae* (*Figure 4C*), consistent with previous

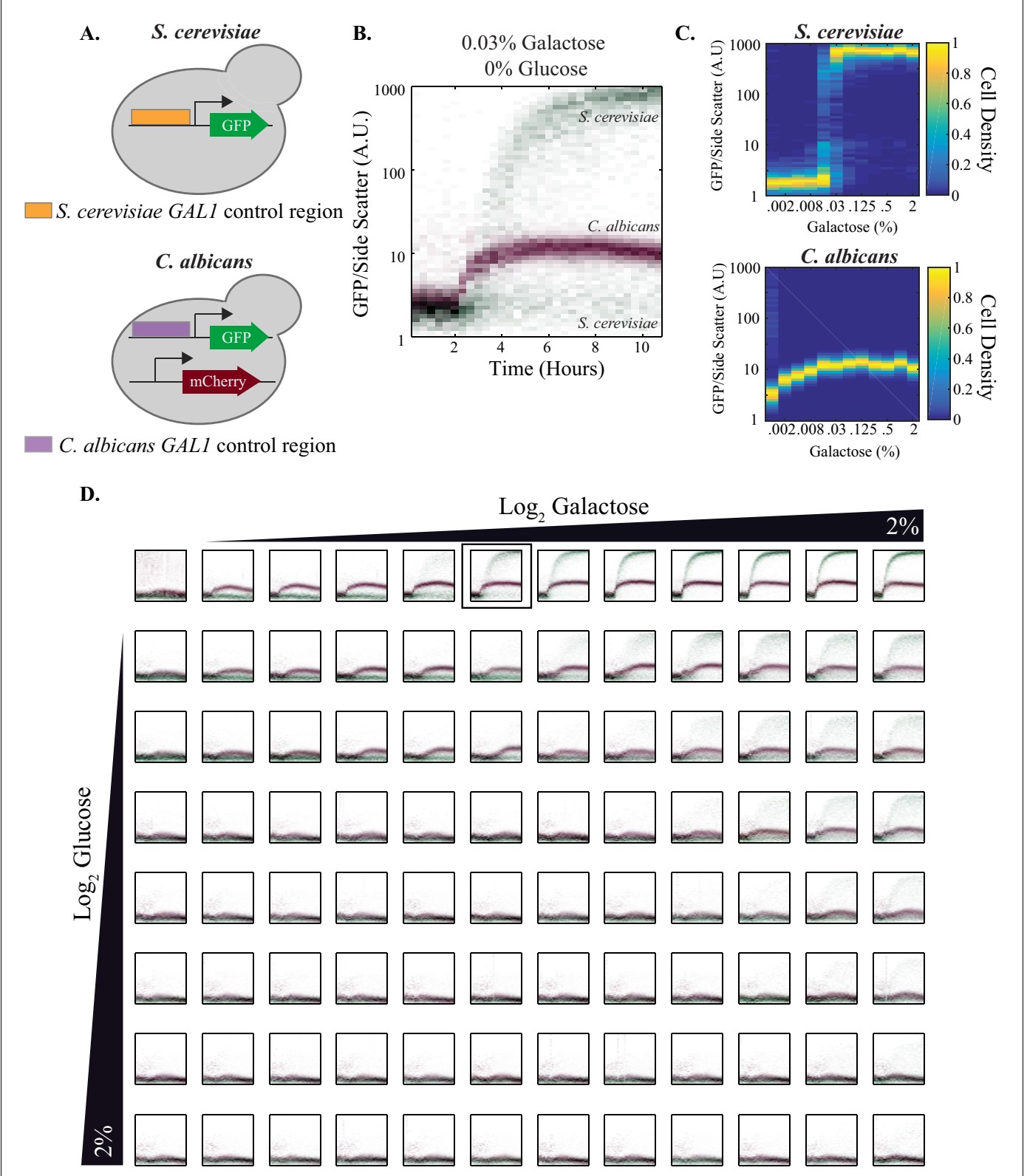

**Figure 4.** Quantitative outputs of *GAL1* gene expression vary between *S. cerevisiae* and *C. albicans*. (**A**) In both species, the *GAL1* promoter was fused to GFP. The *C. albicans* strain also contained a constitutive mCherry reporter that was used to distinguish between the two species. (**B**) The strains were mixed together and fluorescent expression from each species was monitored for ~10 hr. The normalized fluorescence of *GAL1* is plotted as a density-dependent histogram across time. Each dot (red for *C. albicans* and green for *S. cerevisiae*) represents a single cell. The panel shows a timecourse of

*Figure 4 continued on next page*

*Figure 4 continued*

*GAL1* induction for a single concentration of galactose (and with no glucose). (C) Steady-state *GAL1* gene expression of both *S. cerevisiae* (top) and *C. albicans* (bottom) is plotted as a heatmap of cell density. Cell density, the number of cells normalized by the maximum number of cells, represents the fraction of the population at a given expression level. The x-axis represents galactose concentration (in the absence of glucose) while the y- axis represents fluorescent expression. This data has been re-plotted from the panels in row 1 of *Figure 4D*. (D) Behavior of *GAL1* across a wide range of galactose (~2000 fold) and glucose (~128 fold) concentrations, each plotted for ~10 hr. As indicated by the black wedges, galactose concentration increases left to right while glucose concentration increases from top to bottom. Red dots in each plot indicate *C. albicans GAL1* expression while green dots indicate *S. cerevisiae GAL1* expression. The y-axis on each plot is fluorescence, normalized by side scatter, and it spans three orders of magnitude. The data was collected every 20 min for 10 hr. The panel shown in B is indicated by the black square in the top row. See *Figure 4—figure supplement 1* for additional plots from this dataset and *Figure 4—figure supplement 2* for the analysis of *C. albicans* clinical isolates.

The following source data and figure supplements are available for figure 4:

**Source data 1.** Automated flow cytometry data for *S. cerevisiae* and *C. albicans* co-culture experiment across 96 galactose and/or glucose concentrations.
**Source data 2.** Automated flow cytometry data for *C. albicans* SC5314 and other *C. albicans* isolates experiment across galactose concentrations.
**Figure supplement 1.** Quantitative properties of *GAL* gene regulation vary between *S. cerevisiae* and *C. albicans*.
**Figure supplement 2.** Multiple *C. albicans* clinical isolates show similar *GAL1* gene induction in response to galactose.

observations that *C. albicans* is able to respond to very low concentrations of sugar (*Rodaki et al., 2009*). (4) The opposite behavior is observed in the sensitivity of each species to glucose. In the presence of saturating concentrations of galactose, half-maximal repression by glucose occurs at lower levels in *S. cerevisiae* than in *C. albicans* (*Figure 4—figure supplement 1*). (5) A key property of the *S. cerevisiae* network is bimodal expression of p*GAL1*; that is, at intermediate levels of galactose and glucose, the population consists of mixtures of fully ON and fully OFF cells (*Biggar and Crabtree, 2001*; *Venturelli et al., 2015*). No bimodality is evident in *C. albicans* across a large range of glucose and galactose ratios (*Figure 4B–D*, *Figure 4—figure supplement 1*). Instead, the *C. albicans* response appears graded; that is, at intermediate concentrations of glucose and galactose, individual cells produced intermediate levels of GFP (*Figure 4C*).

In comparing the physiologic response of the *GAL* genes (or any other genes for that matter) between different species, it is important to have some knowledge of the variation in response among individuals of the same species. Several groups have shown that the expression dynamics of *GAL1*, specifically its potent induction and bimodality, is conserved across many isolates of *S. cerevisiae* (*Nogi, 1986*; *Wang et al., 2015*; *Warringer et al., 2011*), including W303, the strain used in our analysis (*Ralser et al., 2012*). To determine whether different *C. albicans* isolates vary in their response to galactose, we measured *GAL1* expression dynamics using 11 different patient isolates of *C. albicans,* all from different clades (*Blignaut et al., 2002*; *Lockhart et al., 1996*; *Odds et al., 2007*; *Pujol et al., 2002*), and including strains isolated from different anatomical sites of infection and different parts of the world (*Angebault et al., 2013*; *Hirakawa et al., 2015*; *Odds et al., 2007*; *Shin et al., 2011*; *Wu et al., 2007*, *Supplementary file 1*). Each isolate was engineered so that the *GAL1* promoter was fused to GFP; the isolates were then analyzed across a wide range of galactose concentrations with a mCherry-marked SC5314 strain of *C. albicans* as a control (*Figure 4—figure supplement 2*). We observed that, despite some small differences (*Figure 4—figure supplement 2*), the qualitative and quantitative aspects of *GAL1* induction were similar across all 12 isolates, including SC5314, the lab strain used for most of our experiments (*Figure 4—figure supplement 2*). None of the five differences documented between *S. cerevisiae* and *C. albicans* were observed between any two *C. albicans* isolates (*Figure 4—figure supplement 2*). We conclude from these experiments that the transcriptional response of *S. cerevisiae* and *C. albicans* to galactose—although similar in overall logic—differ in almost all of their quantitative output features. Moreover, these differences are characteristic of each species as a whole and not of a particular isolate.

## Comparing the genes induced by galactose in *C. albicans* to those induced in *S. cerevisiae*

We next tested whether the rewiring of the *GAL* genes had consequences for the regulons that these genes are part of. We compared, by RNA-sequencing, the genes induced in *S. cerevisiae* and *C. albicans* by galactose and glucose compared to a medium lacking a sugar (*Figure 5*, *Figure 5—figure supplements 1* and *2*). In *S. cerevisiae*, consistent with previous work, we observed very high (>275-fold) galactose-induced expression of the three genes encoding the Leloir enzymes (*GAL1, GAL7, GAL10)* as well as that encoding the galactose permease *GAL2* (*Figure 5A*). In *C. albicans* [which lacks a *GAL2* ortholog (*Brown et al, 2009*)]*, GAL1, GAL7* and *GAL10* were all induced by galactose, but to a much lesser extent than in *S. cerevisiae* (*Figure 5B*). As discussed above, this lower induction ratio is due to both higher basal expression (expression in media lacking a sugar) and lower induced levels (in galactose). These observations are all consistent with the single-cell measurements described above; they also show that the p*GAL1* reporter is a good proxy, in both species, for the *GAL* genes in general.

Next, we used this RNA-seq data to determine whether the complete set of genes induced by galactose differs between *S. cerevisiae* and *C. albicans*. In *S. cerevisiae*, nearly all (28/30) genes besides *GAL1, GAL7, GAL10* and *GAL2* are induced by galactose, but to a much lesser extent (at least two-fold but less than 30-fold in two independent experiments, with p-values <0.01). Most of these 30 genes are involved in some aspect of carbohydrate metabolism (*Cherry et al., 2012*, *Supplementary file 2*). In *C. albicans*, 33 genes met the same criteria for being galactose-induced, yet none of them showed the higher levels of induction characteristic of *S. cerevisiae GAL* genes; the majority were induced between two-fold and 13-fold. These genes are annotated as being involved in pathogenesis, biofilm formation, filamentous growth, as well as carbohydrate metabolism (*Inglis et al., 2012*, *Supplementary file 3*). Except for the *GAL1, GAL7* and *GAL10* genes, there was little overlap (only 2 additional genes, *RHR2* and *PFK27*, annotated in *S. cerevisiae* as enzymes involved in sugar metabolism) between the *S. cerevisiae* and *C. albicans* galactose-induced genes (*Figure 5—figure supplement 2*, *Supplementary file 3*). Indeed, many of the galactose-induced genes in *C. albicans* did not have an identifiable ortholog in *S. cerevisiae* and vice-versa (*Byrne and Wolfe, 2005*; *Fitzpatrick et al., 2010*; *Maguire et al., 2013*, *Figure 5—figure supplement 2*). These observations indicate that the wiring change between *S. cerevisiae* and *C. albicans* maintained the galactose induction of *GAL1, GAL7*, and *GAL10* but changed the remaining genes in the galactose-induced regulon.

## Comparing the signals used to induce *GAL* genes in *C. albicans* to those used to induce the *GAL* genes in *S. cerevisiae*

The inclusion of *GAL1, GAL7* and *GAL10* in a larger regulon controlled by Rtg1 and Rtg3 suggests that signals besides galactose may induce the regulon in *C. albicans*. Consistent with this idea, the *C. albicans GAL* genes were observed, in a genome-wide study, to be induced in response to N-acetylglucosamine (GlcNAc), a sugar derivative also known to induce genes involved in pathogenesis, biofilm formation and filamentous growth (*Gunasekera et al., 2010*; *Kamthan et al., 2013*). To further examine this observation, we measured the expression of the *GAL1* promoter fusion in both *C. albicans* and *S. cerevisiae* in response to GlcNAc and observed that GlcNAc induced expression of the *GAL* genes in *C. albicans*, but not *S. cerevisiae* (*Figure 6A*). Moreover, this induction is also observed in our test construct, which contained two Rtg1-Rtg3 binding sites driving expression of a heterologous promoter (*Figure 6B*). The observed GlcNAc induction of both the artificial constructs and the natural *GAL1* promoter requires Rtg1-Rtg3 (*Figure 6A–B*). While the *GAL* genes are induced in response to GlcNAc, they are not required for growth on GlcNAc as a sole sugar source (*Figure 6—figure supplement 1*).

From these experiments, we conclude that, as a direct result of transcriptional rewiring, the regulon to which the *GAL1, GAL7*, and *GAL10* genes belong differs substantially between *S. cerevisiae* and *C. albicans*. Moreover, the regulation of three Leloir enzymes has under gone an important qualitative change; in *C. albicans*, they can be induced by sugars other than galactose.

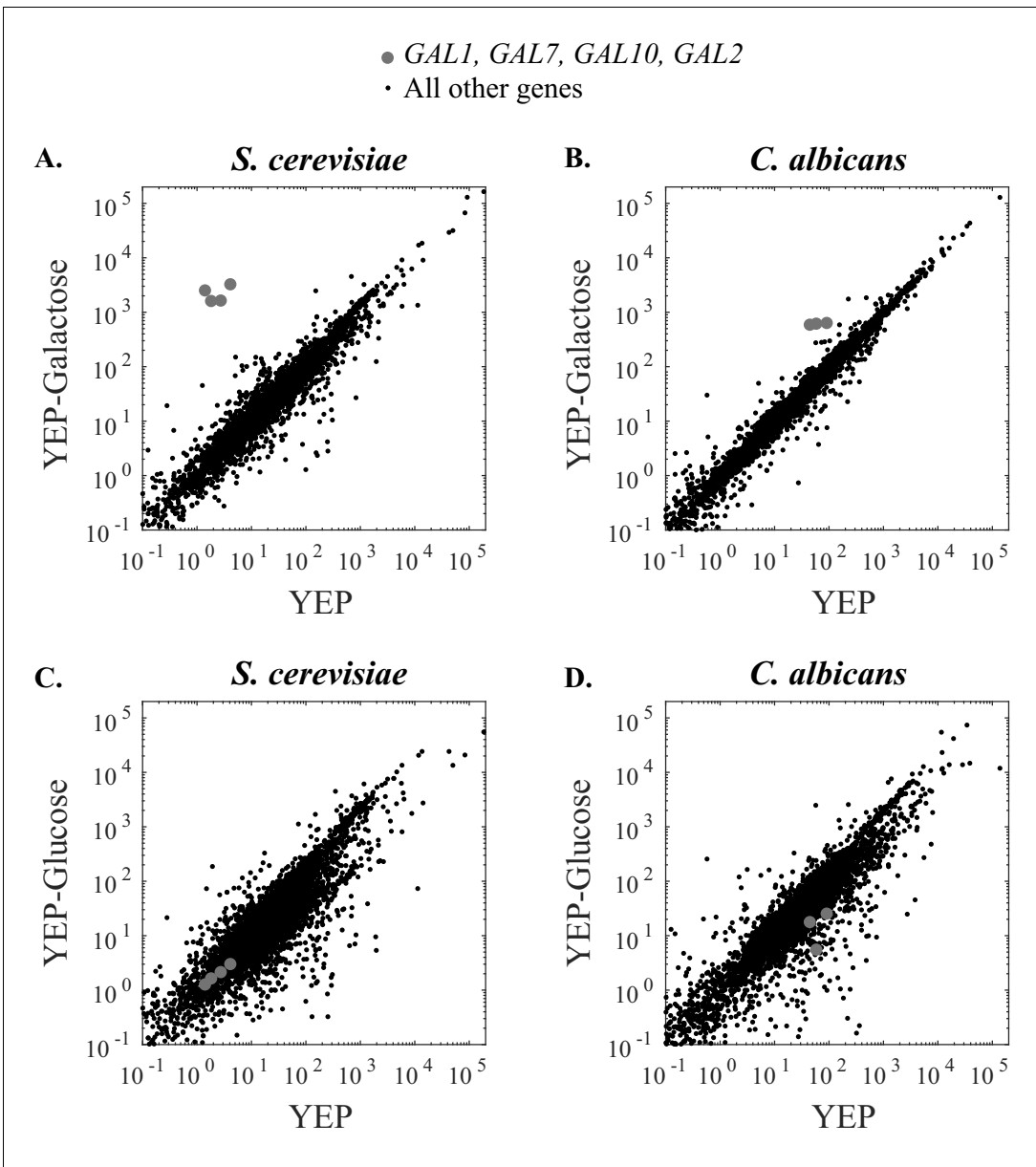

**Figure 5.** The strength of galactose induction differs between *S. cerevisiae and C. albicans*. (**A**) RNA Sequence reads for all genes in *S. cerevisiae* are plotted over two conditions. The x- axis indicates the number of transcripts per million reads when cells were grown in YEP media for 6 hr while the y-axis indicates the number of transcripts per million reads when cells were grown in YEP + 2% galactose for 6 hr. The *GAL1, GAL7, GAL10,* and *GAL2* genes are highly induced in galactose and indicated by grey dots while all other genes are shown in black. (**B**) RNA Sequence reads for all genes in *C. albicans* are plotted across the same two conditions. The *GAL1, GAL7,* and *GAL10* genes are shown in grey (*C. albicans* lacks a *GAL2* ortholog). (**C** and **D**) Here we show the same basic experiment, except that the comparison is between YEP and YEP + 2% glucose, rather than galactose. Additional RNA-sequencing data are provided in *Figure 5—figure supplement 1*.

The following source data and figure supplements are available for figure 5:

**Source data 1.** Quantitative analysis of *S. cerevisiae* RNA-Sequencing data.

**Source data 2.** Quantitative analysis of *C. albicans* RNA-Sequencing data.

**Figure supplement 1.** RNA-Sequencing of *S. cerevisiae* and *C. albicans* replicates in different sugar sources (glucose, no sugar, galactose).

*Figure 5 continued on next page*

*Figure 5 continued*

**Figure supplement 2.** Genes induced by galactose show little overlap between *S. cerevisiae* and *C. albicans*.

## Inferring the evolutionary transition of *GAL* regulation

So far, we have documented a wiring difference in *GAL1*, *GAL7*, and *GAL10* gene regulation between the *Saccharomyces* and *Candida* clades; here we address when the change occurred. Specifically, we asked whether Gal4 or Rtg1-Rtg3 was the regulator of these genes in the ancestor of *C. albicans* and *S. cerevisiae*. To determine this, we examined *Yarrowia lipolytica,* an outgroup species, that is, a species that branched before the *Candida* and *Saccharomycotina* clades diverged (*Figure 7A*). The *Y. lipolytica* genome contains clear orthologs for Gal1, Gal7 and Gal10. Moreover, the ability of *Y. lipolytica* to metabolize galactose increases when these genes are overexpressed (*Lazar et al., 2015*), indicating that they carry out the same role as the orthologous genes in *S. cerevisiae* and *C. albicans*. *Y. lipolytica* lacks a clear ortholog for Gal4, the closest relatives being other zinc cluster proteins $(Zn(II)_2Cys_6)$ lacking the surrounding amino acid sequences characteristic of the Gal4 orthologs of many other fungal species. The *Y. lipolytica* genome does contain a single *RTG* ortholog (*YALI0F11979*); it is more similar to Rtg1 than Rtg3, and we will refer to the *Y. lipolytica* gene as *RTG1*.

We knocked out the *RTG1* gene in *Y. lipolytica* and measured the mRNA expression of *GAL1* in cells grown in galactose. We found that in a *Y. lipolytica* parent strain, galactose induces *GAL1* expression at least three-fold. This induction is not observed in two independently constructed *rtg1* knockout strains (*Figure 7b*), indicating that this transcriptional regulator plays an important role in activating *GAL1* in *Y. lipolytica*. We note that knocking out this regulator did not result in a growth defect of *Y. lipolytica* on 2% galactose (*Figure 7—figure supplement 1*). However, the *GAL* genes in this species appear to be expressed at a relatively high basal level.

Taken together, all the phylogenetic results indicate that the Gal4 mode of *GAL* gene regulation likely arose along the *S. cerevisiae* lineage after *S. cerevisiae* and *C. albicans* diverged (*Figure 7C*). We also know that it occurred before *S. cerevisiae* and *K. lactis* diverged as *K. lactis* also uses the Gal4 mode of regulation (*Rubio-Texeira, 2005*). Finally, we know from recent work (*Roop et al.,*

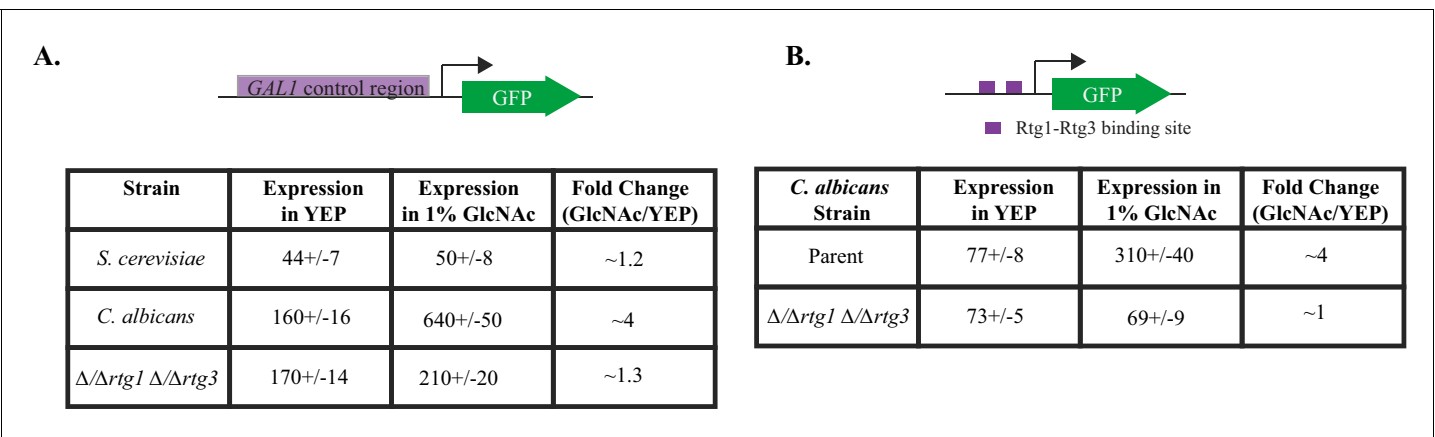

**Figure 6.** GlcNAc induces *GAL* gene expression in *C. albicans* through Rtg1-Rtg3. (**A**) *GAL1-* GFP expression in *C. albicans* and *S. cerevisiae* was measured in cells grown in YEP +1% GlcNAc for 6 hr, compared to expression in YEP. (**B**) GFP expression from a *C. albicans* strain containing the test construct where the Rtg1-Rtg3 *cis*-regulatory motifs are driving a heterologous promoter was measured in similar conditions. Mean expression levels are reported in arbitrary units. Errors indicate standard errors derived from three independent measurements.

The following figure supplement is available for figure 6:

**Figure supplement 1.** The *GAL* genes do not contribute to growth on GlcNAc.

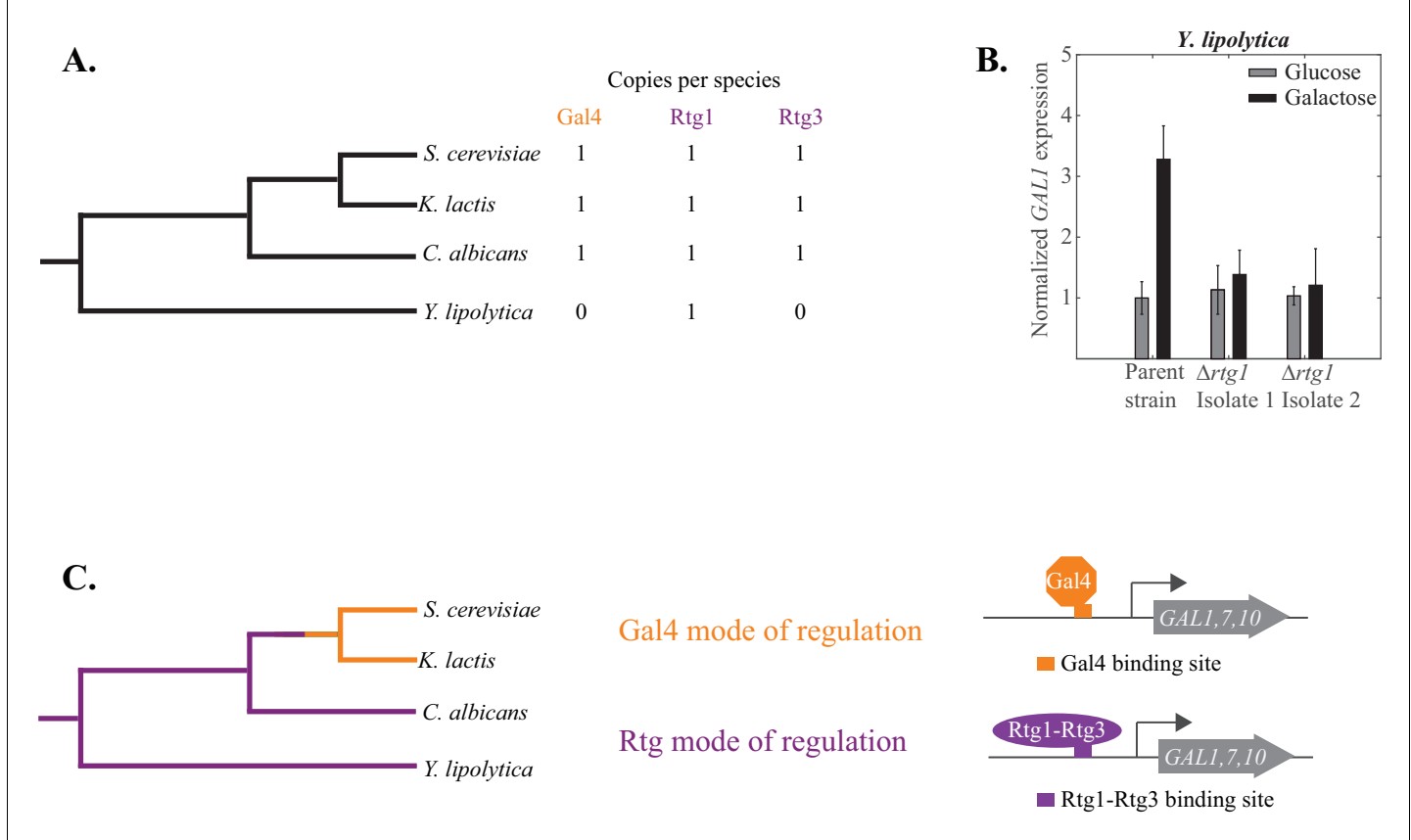

**Figure 7.** The Rtg mode of galactose-induction is ancestral to the Gal4 mode. (**A**) *Y. lipolytica* branched before the *S. cerevisiae* and *C. albicans* clades diverged from each other. Its genome lacks an identifiable Gal4 ortholog but contains a clear Rtg1 ortholog. (**B**). mRNA expression of *GAL1* was measured by qPCR in glucose and galactose in a *Y. lipolytica* parent strain and two independently constructed strains where the Rtg1 ortholog (*YALI0F11979*) was deleted. The relative expression level of the parent strain in glucose (compared to *SGA1*, a housekeeping gene) was normalized to 1 and expression of all other strains and conditions were plotted relative to it. (**C**) The results suggest that the Rtg mode of regulation (purple line) is ancestral to the Gal4 mode (orange line), which appeared before *S. cerevisiae* and *K. lactis* diverged from each other.

The following figure supplement is available for figure 7:

**Figure supplement 1.** The *Y. lipolytica* Rtg1 ortholog does not appear to contribute to the growth of *Y. lipolytica* on galactose.

2016) that the tight repression of *GAL1*, *GAL7*, and *GAL10* by glucose–a defining characteristic of *S. cerevisiae*–occurred much later than the Rtg1-Rtg3 to Gal4 rewiring event, specifically after *S. cerevisiae* branched from *S. paradoxus*.

## Discussion

The regulation of central carbon metabolism is crucial for all species (*Sandai et al., 2012*). Here, we documented an evolutionary shift in the molecular mechanisms through which the enzymes that metabolize galactose (Gal1, Gal7, Gal10) are specifically induced by galactose. These enzymes and their functions are conserved across all kingdoms of life; direct evidence shows this is the case for many fungal species including *S. cerevisiae* (*Dudley et al., 2005*), *K. lactis* (*Rubio-Texeira, 2005*), *C. albicans* (*Martchenko et al., 2007a*, this paper) and *Y. lipolytica* (*Lazar et al., 2015*). These enzymes have a deeply conserved function (conversion of galactose to glucose-1 phosphate) and a deeply conserved pattern of regulation: their expression increases when galactose is added to the growth medium. However, a shift in the mechanism of galactose induction occurred along the evolutionary pathway to *S. cerevisiae* resulting in Gal4 'replacing' Rtg1-Rtg3 as an inducer of the *GAL* genes in

response to galactose. This ancestral mode, the regulation of the *GAL* genes by Rtg1-Rtg3, was retained in the *C. albicans* clade.

*S. cerevisiae* and *C. albicans* both have clear orthologs of Gal4, Rtg1, and Rtg3 (*Fitzpatrick et al., 2010*; *Wapinski et al., 2007*), and these transcriptional regulators have retained their DNA-binding specificities since the two clades branched, at least 300 million years ago (*Askew et al., 2009*; *Pérez et al., 2013*; *Taylor and Berbee, 2006*, this paper). The rewiring therefore occurred, at least in large part, through changes in the *cis*-regulatory sequences of the control regions of the *GAL* genes. In *C. albicans*, we show that the *cis*-regulatory sequences for Rtg1-Rtg3, located next to the *GAL* genes, are sufficient for galactose induction; in the *S. cerevisiae* clade, these were 'replaced' by *cis*-regulatory sequences for Gal4.

To understand the consequences of this wiring change, we directly compared, using fluorescent reporters combined with a robotically controlled flow cytometer, the quantitative responses of *C. albicans* and *S. cerevisiae* to wide ranges of galactose and glucose concentrations. We documented five important differences between the *C. albicans* and *S. cerevisiae* galactose responses. (1) The basal level of *GAL* gene expression is higher in *C. albicans* and the maximal induced level is lower resulting in a significantly lower induction ratio (~12-fold) in *C. albicans* compared with *S. cerevisiae* (~900-fold). (2) Induction occurs 20–40 min faster in *C. albicans* than in *S. cerevisiae*, with the exact difference dependent on the medium composition. (3) The concentration of galactose required for half maximal induction of *GAL1* is 15-fold lower in *C. albicans* (0.002%) than *S. cerevisiae* (0.03%). (4) The concentration of glucose required for half-maximal repression is at least two-fold lower in *S. cerevisiae* than in *C. albicans*. (5) The well-documented bimodality observed at certain ratios of glucose: galactose in *S. cerevisiae* appears absent in *C. albicans* over a wide range of glucose and galactose concentrations; thus, instead of an all-or-none response, *C. albicans* exhibits a much more graded expression. We showed that these characteristics were true for 11 different clinical isolates of *C. albicans*, demonstrating that these behaviors are characteristic of the species as whole rather than a particular isolate. Full genome analysis revealed three additional, qualitative differences between regulation of the *GAL* genes in *C. albicans* and *S. cerevisiae*. (6) In *C. albicans*, the galactose regulon (33 genes induced by galactose) includes, in addition to the *GAL* genes, genes implicated in several aspects of pathogenesis. In contrast, the galactose regulon of *S. cerevisiae* functions almost exclusively in galactose regulation, metabolism and transport. Other than *GAL1*, *GAL7* and *GAL10*, there is virtually no overlap (only two genes) between the galactose-induced regulon in *S. cerevisiae* and *C. albicans*. (7) The galactose regulon of *S. cerevisiae* consists of two major tiers of regulation: Four *GAL* genes that are induced nearly 1000 fold in response to galactose and 28 genes that are induced to a much lesser extent (2 to 30 fold, *Supplementary file 3*). In *C. albicans*, 33 genes that are induced by galactose all show modest induction (31 are induced 2 to 13 fold while the other 2 are induced 20 and 50 fold, respectively). (8) In *C. albicans*, the *GAL* genes (and the rest of the regulon) can be induced by signals other than galactose.

Although we cannot pinpoint the extent to which the Rtg1-Rtg3 to Gal4 rewiring contributes to each of these eight differences, we can say that at least two of them (the induction ratios of the *GAL* genes and the response of the *GAL* genes to non-galactose signals) critically depend on the rewiring. The moderate induction by galactose of the *GAL* genes in *C. albicans* is recapitulated when Rtg1-Rtg3 *cis*-regulatory sequences are added to a test promoter. Likewise, the induction by GlcNAc in *C. albicans* is recapitulated in a test construct containing Rtg1-Rtg3 sites and is destroyed when *RTG1* and *RTG3* are deleted in *C. albicans*.

Although there is no direct evidence that this rewiring was adaptive, it is tempting to speculate that the characteristics of galactose induction in *S. cerevisiae*—high induction of *GAL* genes, low sensitivity to galactose concentration, high sensitivity to glucose concentration, and specificity for galactose—are advantageous for a species that ferments different sugars rapidly (*Johnston, 1999*). In contrast, the features of *C. albicans* (high basal levels and lower induction of *GAL* gene expression, higher sensitivity to galactose, and the inclusion of the *GAL* genes in a regulon that includes genes needed for pathogenesis) may be important for *C. albicans* to thrive in niches of warm-blooded animals where galactose is probably present at very low concentrations (*Gibson et al., 1996*). For example, in many mammals, galactose is synthesized and incorporated into glycolipids and glycoproteins; it is ingested through diet, for example, through the hydrolysis of lactose into glucose and galactose. Consistent with these ideas, we know that the Rtg1-Rtg3 circuit is central to the ability of *C. albicans* to proliferate in a mammalian host, as *C. albicans* mutants deleted for *RTG1*

and *RTG3* are deficient in several mouse models of infection (*Pérez et al., 2013*). Although we found that the Rtg1-Rtg3 mode of *GAL* gene regulation is ancestral, this does not mean that the ancestral fungal species resembled *C. albicans* in its ability to thrive in animal hosts. The Rtg1-Rtg3 circuit continued to evolve in the *C. albicans* clade and it seems likely that additional changes in this circuitry were needed to adapt specifically for life in a mammalian host.

In *C. albicans*, why does galactose induce genes involved in pathogenesis? We suggest two possibilities for this observation. First, galactose may act as a proxy for a type of environment frequently encountered by *C. albicans* in the host. According to this idea, the presence of low levels of galactose is a signal for *C. albicans* to mount a general response that involves both metabolizing galactose and inducing virulence properties. This 'proxy' idea might also explain why other sugars also induce this regulon in *C. albicans*. A second possibility is that galactose is utilized differently in *C. albicans* than in *S. cerevisiae*; rather than simply converting galactose to glucose-1 phosphate, *C. albicans* may use endogenous galactose (or metabolites derived from galactose) in ways that require the induction of other genes.

Irrespective of the possible evolutionary advantages of the Gal4 mode of regulation in *S. cerevisiae* versus the Rtg1-Rtg3 mode of regulation in *C. albicans*, the work presented here shows that the regulatory schemes produce different circuit characteristics. Although the overall logic of *GAL* gene regulation remains the same (the *GAL* genes are induced by galactose in both species), almost all other features of the circuit differ significantly, ranging from quantitative output (kinetics, induction ratio, bimodality) to the structure of the regulons that include the *GAL* genes in *C. albicans*. This study illustrates how transcriptional rewiring over evolutionary timescales can preserve a basic circuit output yet alter almost all of its quantitative and qualitative features.

## Materials and methods

### Orthology analysis

The presence of orthologs for Gal1, Gal7 and Gal10 had previously been established in both *C. albicans* and *Y. lipolytica* (*Slot and Rokas, 2010*). The orthology of other genes (Gal80 in *C. albicans*) and Rtg1 in *Y. lipolytica* were determined from Fungal Orthogroups (*Wapinski et al., 2007*). BLASTP searches (*Altschul et al., 1990*) conducted with protein sequences from *Candida albicans* did not find strong sequence similarity to Gal4 in *Y. lipolytica*; the best 'hits' in each of these species consist of short amino acid segments. None of these 'hits' contain the cysteine residues that are characteristic of the $Zn(II)_2/Cys_6$ class of proteins that Gal4 belongs to. As such, we do not consider them orthologs.

### Plasmid construction

The GFP tagging plasmid template pMBL179, a gift from Dr. Matthew B. Lohse, was constructed as follows. The *C. albicans* optimized GFP sequence (*Cormack et al., 1997*) was inserted into pUC19 between the HindIII and PstI sites. The *SAT1* selectable marker from pNIM1 (*Park and Morschhäuser, 2005*) was amplified and then inserted between the PstI and BamHI sites. The 308 nucleotides just before the *GAL1* gene were amplified from *C. albicans* SC5314 genomic DNA and inserted in the HindIII site of pMBL179. Then the 350 nucleotides just after the *GAL1* start codon were amplified from *C. albicans* genomic DNA and inserted into EcoRI and BamHI sites of the resulting plasmid. The final plasmid, pCKD004, was linearized with BspH1 and EcoRI and transformed into *C. albicans* strains to make a *GAL1*-GFP promoter fusion construct.

The synthetic reporter construct, pCKD017, was constructed by amplifying the 600 nucleotides just before the translational start site of *CYC1* from *C. albicans* SC5314 genomic DNA and inserting it just upstream of GFP in pMBL179. Oligonucleotides containing two putative Rtg1-Rtg3 binding sites were annealed and inserted into XhoI and SphI sites of the resulting plasmid, pLN2, a gift from Liron Noiman. The oligonucleotide sequences are as follows (the binding sites are in bold):

TCGAGGACGTCTGTACAAA**AATGTAACGTTAC**ATTAAGATTAA**AATGTAACGTTAC**AAATTCCA TCTTTATACCATGGGCATG

CCTGCAGACATGTTTTTA**CATTGCAATGTAA**TTCTAATTTTA**CATTGCAATGTTTAA**GGTAGAAA TATGGTACCC

The plasmid used to tag Rpl26b with mCherry, pMBL186, was constructed as follows. 500 base pairs upstream and downstream of the Rpl26b stop codon were amplified from *C. albicans* SC5314 genomic DNA. The mCherry DNA sequence (*Shaner et al., 2004*) was modified to account for *C. albicans* codon usage and alternative coding of CTG (*Lloyd and Sharp, 1992*) and synthesized by DNA 2.0 (*Lohse and Johnson, 2016*). The 500 base pairs upstream of the Rpl26b stop codon, mCherry and the 500 base pairs downstream of the Rpl26b stop codon were fused together with PCR and inserted into the SphI and AatII sites of pUC19 to create pMBL182. pSFS2a (*Reuss et al., 2004*) was digested with XhoI and NotI and the sequences encoding the recyclable *SATI* marker were subcloned into pMBL182, creating pMBL186. pMBL186 was linearized with SphI and AatII and transformed into AHY135 (*Lohse et al., 2013*).

pCKD016, the plasmid used to knockout *YALI0F11979*, the *RTG1* ortholog in *Y. lipolytica*, was constructed by serially cloning in the flanking sequences of *RTG1* into pFA6a-hbh-hphmx4 (*Tagwerker et al., 2006*). The 700 base pairs upstream of the coding region of *RTG1* were amplified from *Y. lipolytica* genomic DNA (P01 strain, a gift from Claude Gaillardin) and inserted into the BamH1 and EcoR1 sites of pFA6a-hbh-hphmx4. The 700 base pairs downstream of the coding region of *RTG1* were similarly amplified and inserted into the SpeI and ClaI sites of the resulting plasmid. The final plasmid, pCKD016, was linearized with SpeI and HindIII, purified with a PCR purification kit (Qiagen, Valencia, CA) and transformed into *Y. lipolytica* to make a *RTG1* knockout strain.

## Strain construction

### *C. albicans* knockout strains

All knockout strains were derived from SN152 (*Noble and Johnson, 2005*) and constructed by fusion PCR using the His and Leu cassettes as previously described (*Hernday et al., 2010*; *Homann et al., 2009*; *Noble and Johnson, 2005*). OH13 (*Homann et al., 2009*) was used as a parent strain. The knockout strains used in the *in vivo* rat catheter model were made Arg+ by transforming individual knockouts with PmeI-digested pSN105 (*Noble et al., 2010*). In this case, SN425 (*Noble et al., 2010*) was used as a parent strain.

### *C. albicans* fluorescent reporter strains

Transcriptional reporter strains of *GAL1* were constructed by transforming linearized pCKD004 into wild-type and knockout strains. Transformants were selected on YPD + 400 µg/ml clonNAT plates (*Nobile et al., 2008*). 1 liter of these plates contains 20 g Bacto-Agar, 20 g Bacto-Peptone, 10 g Yeast Extract, 0.4 g cloNat, 50 ml 40% glucose and 950 ml water. The Rpl26b-mCherry strain was constructed by transforming linearized pMBL186 into wild-type strains. Transformants were selected onto YPD + 400 µg/ml clonNAT plates. The *SAT1* marker was recycled as described previously (*Reuss et al., 2004*).

### *Y. lipolytica* knockouts

The construction of knockout strains in *Y. lipolytica* was adapted from Davidow et al (*Davidow et al., 1987*). 10 ml YPD (20g Bacto-Peptone, 10 g Yeast Extract, 50 ml 40% glucose in 950 ml of water) cultures of the P01 strain of *Y. lipolytica* (CD329), a gift from Professor Claude Gaillardin (*Barth and Gaillardin, 1996*), were pelleted and resuspended in 2 ml of 10 mM Tris, 1 mM EDTA. They were then repelleted and resuspended in ~400 µl of 10 mM Tris, 1 mM EDTA, 0.1 M lithium acetate and mixed gently at 28°C for 1 hr. 100 µl of cells were added to 1 µg linearized pCKD016 and 50 µg boiled salmon sperm DNA. The mixture was incubated at 28°C for 30 min. ~700 µl of 50% PEG 3350, 10 mM Tris, 1 mM EDTA, 0.1 M lithium acetate were added to the transformation tube. After a 1-hr incubation at 28°C, a 5-minute heat shock (37°C) was applied to the transformation tube. Cells were washed, pelleted, resuspended in water and plated onto YPD plates (20 g Bacto-Agar, 20 g Bacto-Peptone, 10 g Yeast Extract, 50 ml 40% glucose in 950 ml of water) and incubated at 25°C. After one day, the YPD plates were replica plated onto YPD + 200 mg/L hygromycin (20 g Bacto-Agar, 20g Bacto-Peptone, 10 g Yeast Extract, 50 ml 40% glucose, 4 ml 50 mg/ml hygromycin B in 950 ml of water). Colonies were verified by diagnostic PCR by attempting to amplify a small internal fragment of the coding region of *RTG1* in *Y. lipolytica* (for a successful deletion, this intra-coding region PCR yielded no product while a wild-type control yielded a clear product) to create strains CD331 and CD333.

### S.cerevisiae strains

A derivative of the w303 strain of *S. cerevisiae* (CD014) was used in *Figure 4*; *Figure 4—figure supplement 1*; *Figure 5*; *Figure 5—figure supplement 1* and *Figure 5—figure supplement 2*. This strain was transformed (*Longtine et al., 1998*) with DNA sequence containing the intergenic region of *GAL1* fused to GFP and the *URA3* marker. This sequence was flanked with homology to *LEU2*. Colonies were selected on media lacking uracil (2% Dextrose, 6.7% Yeast Nitrogen Base with ammonium sulfate and amino acids) and verified by amplifying the flanks of the *LEU2* locus and the GFP itself (strain CD015).

### Plating assays

*C. albicans* strains were grown overnight at 30°C in S-Raffinose (2% raffinose, 6.7% Yeast Nitrogen Base with ammonium sulfate and a full complement of amino acids). Strains were diluted ~100-1000 fold in the morning and grown for 4-6 hr. Strains were serially diluted 10-fold into S-Raffinose five times and ~5 µl of cells from each dilution were spotted onto SD (2% Dextrose, 6.7% Yeast Nitrogen Based with ammonium sulfate and amino acids), S-Galactose (2% Galactose, 6.7% Yeast Nitrogen Based with ammonium sulfate and amino acids) plates ± 3 µg/ml Antimycin A (Millipore Sigma, St. Louis, MO), or S-GlcNAc (2% Galactose, 6.7% Yeast Nitrogen Based with ammonium sulfate and amino acids) plates. Strain growth was monitored visually; strains that grew well exhibited growth across all dilutions. We note that *Y. lipolytica* is an obligate respirator; hence, Antimycin A was not included in any *Y. lipolytica* plating assays.

### Flow cytometry

Cells were grown at 30°C overnight in rich media lacking a sugar source. This media, YEP, contained 10 g/L yeast extract and 20 g/L Bacto-peptone. This culture was diluted to an optical density ($OD_{600}$) of approximately 0.05–0.2 and grown for an additional 6 hr in YEP media ± the indicated sugar (glucose, galactose or GlcNAc, all purchased from Millipore Sigma, St. Louis, MO).

Single-cell fluorescence was measured on a LSRII analyzer (BD Biosciences). A blue (488 nm) laser was used to excite GFP and emission was detected using a 530/30 nm bandpass filter. A yellow-green laser (561 nm) was used to excite mCherry and emission was detected using a 610/20 nm bandpass filter. For each sample, 5,000–30,000 cells were measured and the mean fluorescence level was calculated. Each sample was measured three times; the mean of all three independent measurements is reported in *Figures 1*, *3* and *6*. The errors in these figures refer to the standard error of the mean of these three measurements.

### Automated flow cytometry

Cells were grown at 30°C overnight in YEP, diluted and grown for an additional 6 hr in YEP media. This single well-mixed culture was then diluted with YEP media into a 96-well microtiter plate to a final $OD_{600}$ of less than 0.05 (500 µL volume). Cells were grown in the 96-well plate and diluted every 20 min by a liquid-handling robot with YEP media for ~2 hr prior to induction with glucose and galactose, as described previously (*Venturelli et al., 2015*; *Zuleta et al., 2014*). A 30 µl sample was removed from the culture for measurement on the cytometer at each time point and 30 µl of fresh YEP media containing the appropriate 1X concentration of glucose and galactose was used to maintain a constant culture volume. Single-cell measurements were taken on the LSRII analyzer, as described above. For additional information on the hardware, software and data processing of the automated flow cytometry system, see Zuleta et al (*Zuleta et al., 2014*).

### Motif analysis

Genome sequences of *S. cerevisiae, N. castelli, T. pfaffii, Z. rouxii, L. thermotolerans* and *K. lactis* were concatenated into a single file. Similarly, genome sequences of *C. albicans, C. dubliniensis, C. tropicalis, C. parapsilosis, C. lusitaniae*, and *C. guillermondii* were concatenated into a single file. Motifs in intergenic regions of all orthologs of *GAL1*, *GAL7*, and *GAL10* [as identified by *Byrne and Wolfe (2005)*; *Fitzpatrick et al. (2010a)*; *Maguire et al. (2013)*] in each group were identified using SCOPE (*Carlson et al., 2007*; *Chakravarty et al., 2007*). The top motifs from each group of species are shown in *Figure 3—figure supplement 1*.

## *In vivo* rat catheter biofilm model

The well-established rat central-venous catheter infection model (*Andes et al., 2004*) was used for *in vivo* biofilm modeling to mimic human catheter infections, as described previously (*Andes et al., 2004*; *Nobile et al., 2006*). For this model, specific-pathogen-free female Sprague-Dawley rats weighing 400 grams (Harlan Sprague-Dawley, RRID:RGD_5508397) were used. A heparinized (100 Units/ml) polyethylene catheter with 0.76 mm inner and 1.52 mm outer diameters was inserted into the external jugular vein and advanced to a site above the right atrium. The catheter was secured to the vein with the proximal end tunneled subcutaneously to the midscapular space and externalized through the skin. The catheters were inserted 24 hr prior to infection to permit a conditioning period for deposition of host protein on the catheter surface. Infection was achieved by intraluminal instillation of 500 µl *C. albicans* cells ($10^6$ cells/ml). After a 4 hr dwelling period, the catheter volume was withdrawn and the catheter flushed with heparinized 0.15 M NaCl. Catheters were removed after 24 hr of *C. albicans* infection to assay biofilm development on the intraluminal surface by scanning electron microscopy (SEM). Catheter segments were washed with 0.1 M phosphate buffer, pH 7.2, fixed in 1% glutaraldehyde/4% formaldehyde, washed again with phosphate buffer for 5 min, and placed in 1% osmium tetroxide for 30 min. The samples were dehydrated in a series of 10 min ethanol washes (30%, 50%, 70%, 85%, 95%, and 100%), followed by critical point drying. Specimens were mounted on aluminum stubs, sputter coated with gold, and imaged using a Hitachi S-5700 or JEOL JSM- 6100 scanning electron microscope in the high-vacuum mode at 10 kV. Images were assembled using Adobe Photoshop Version 7.0.1 software. All procedures were approved by the Institutional Animal Care and Use Committee (IACUC) at the University of Wisconsin (protocol MV1947) according to the guidelines of the Animal Welfare Act, and The Institute of Laboratory Animal Resources Guide for the Care and Use of Laboratory Animals and Public Health Service Policy.

## RNA-sequencing and analysis

Two independent sets of samples were grown, processed, and sequenced in both *S. cerevisiae* and *C. albicans*. Samples were harvested as follows: Cells were grown at 30°C overnight in YEP, diluted to $OD_{600} = 0.067$ and grown for an additional 6 hr in YEP, YEP + 2% glucose, or YEP + 2% galactose. Cells were pelleted and RNA was extracted from these pellets using the Ambion RiboPure™ Yeast Kit with the DNase I treatment step. RNA concentration and integrity were assessed with a Bioanalyzer. RNA samples were sent to the Columbia Sulzberger Genome Center where libraries for sequencing were prepared using the TruSeq RNA Library Preparation kit v2 (Illumina, San Diego, CA). Samples were pooled and sequenced (100 bp, single end reads) on a HiSeq 2500 (Illumina, San Diego, CA). Each sample yielded ~30 million reads. The data have been submitted to the sequence read archive (SRA) as accession numbers SRP083773 (*S. cerevisiae*) and SRP083777 (*C. albicans*).

Sequences were pseudo-aligned to the *S. cerevisiae* and *C. albicans* genomes and tabulated using kallisto (*Bray et al., 2016*). Raw data (in transcripts per million reads) are plotted in *Figure 5* and *Figure 5—figure supplement 1* and are available as *Figure 5—source data 1* and *Figure 5—source data 2*. The p-values across conditions (galactose vs YEP) were calculated using sleuth (*Pimentel et al., 2016*), incorporating variance across independent sets of samples as well as across 100 bootstraps of kallisto output.

## q-PCR

25°C overnights of *Y. lipolytica* strains were diluted back to $OD_{600} = 0.2$ in the morning and allowed to regrow for 6 hr in glucose and galactose. The equivalent of 10 mL at $OD_{600} = 1$ was harvested for each culture when cells were pelleted.

RNA was extracted from pellets using the Ambion RiboPure Yeast Kit with the DNase I treatment step. Superscript II RT was used for cDNA synthesis. We used a Power SYBR Green mix (Thermo-Fisher, Waltham, MA) for all qPCR reactions. The qPCR cycle consisted of a 10 min holding step at 95°C, followed by 40 cycles at 95°C for 15 s and 60°C for 1 min. This was followed by a dissociation curve analysis. Three +RT and one –RT reaction were run for each strain, 5 µL of a 1:100 dilution were used in each well. A 1:10 dilution of an equal volume mixture of all +RT reactions was used as the starting point for the standard curve, which consisted of 6 1:4 dilutions from the starting sample. *GAL1* was amplified with the following primers: Forward: GATTTTGCTCCAACCCTCAAG and Reverse: ACCCGCAGATTGTAGTTTCG. *SGA1* was used as a control (*Teste et al., 2009*). *SGA1* was

amplified with the following primers: Forward: ACAATGGAGATATGTCGGAGC and Reverse: TCCC TTTGATAACTTCCTGGC.

## Acknowledgements

We are grateful to Richard Bennett, Marie-Elisabeth Bougnoux and Christophe d'Enfert for clinical isolates of *Candida albicans*, Claude Gaillardin for the P01a strain of *Y. lipolytica*, Alistair Brown for a *GCN4* knockout strain of *C. albicans* (GTC43), Robert Gross for technical assistance with the motif analysis, and the Columbia Sulzberger Genome Center for conducting the RNA-Sequencing. We thank Chris Baker, Scott Coyle, Nairi Hartooni, Joe Levine, Matt Lohse and Mark Ptashne for comments on the manuscript, Ananda Mendoza for technical assistance and previous and current lab members for strains and plasmids; in particular, Oliver Homann, Emily Fox, Matt Lohse, Clarissa Nobile and Julie Takagi for TFKO strains, Liron Noiman for pLN2, a plasmid containing the *C. albicans* test promoter, Matt Lohse for pMBL179, the plasmid containing GFP, and *Candida albicans* strains containing Rpl26b-mCherry and especially Jose C Pérez for knockout strains of Gal1 and Gal10, 'addback' strains of Rtg1 and Rtg3 and for sharing ChIP-chip data. This study was supported by the UCSF Center for Systems and Synthetic Biology (NIGMS P50 GM081879), the Paul G Allen Family Foundation (HES) and NIH grants R01AI073289 (DRA) and R01AI049187 (ADJ).

## Additional information

### Funding

| Funder | Grant reference number | Author |
|---|---|---|
| Paul G. Allen Family Foundation | | Ignacio A Zuleta<br>Hana El-Samad |
| National Institute of General Medical Sciences | P50 GM081879 | Ignacio A Zuleta<br>Hana El-Samad |
| National Institutes of Health | R01AI073289 | Kaitlin F Mitchell<br>David R Andes |
| National Institutes of Health | R01AI049187 | Chiraj K Dalal<br>Alexander D Johnson |

The funders had no role in study design, data collection and interpretation, or the decision to submit the work for publication.

### Author contributions

CKD, IAZ, HE-S, ADJ, Conception and design, Acquisition of data, Analysis and interpretation of data, Drafting or revising the article; KFM, DRA, Conception and design, Acquisition of data, Analysis and interpretation of data

### Author ORCIDs

Chiraj K Dalal, http://orcid.org/0000-0002-3624-8409

### Ethics

Animal experimentation: All procedures in this study were approved by the Institutional Animal Care and Use Committee (IACUC) at the University of Wisconsin (protocol number MV1947) according to the guidelines of the Animal Welfare Act, and The Institute of Laboratory Animal Resources Guide for the Care and Use of Laboratory Animals and Public Health Service Policy.

## Additional files

### Supplementary files

• Supplementary file 1. Related to *Figure 4—figure supplement 2*: Origins of *C. albicans* clinical isolates. The clades, anatomical sites of infection, geographical sites of isolation and types of infection are listed for each isolate.

• Supplementary file 2. Related to *Figure 5—figure supplement 2*: List of genes induced by galactose at least two-fold in independent RNA-sequencing experiments in *S. cerevisiae* as well as gene ontology and orthology analyses of this gene set.

• Supplementary file 3. Related to *Figure 5—figure supplement 2*: List of genes induced by galactose at least two- fold in independent RNA-sequencing experiments in *C. albicans* as well as gene ontology and orthology analyses of this gene set.

• Supplementary file 4. *C. albicans* strains used in this study.

### Major datasets

The following datasets were generated:

| Author(s) | Year | Dataset title | Dataset URL | Database, license, and accessibility information |
|---|---|---|---|---|
| Dalal CK, Zuleta IA, Mitchell KF, Andes DR, El- Samad H, Johnson AD | 2016 | Transcriptional Response of S. cerevisiae to Glucose and Galactose | http://www.ncbi.nlm.nih. gov/sra/SRP083773 | Publicly available at the NCBI Short Read Archive (accession no: SRP083773) |
| Dalal CK, Zuleta IA, Mitchell KF, Andes DR, El-Samad H, Johnson AD | 2016 | Transcriptional Response of C. albicans to Glucose and Galactose | http://www.ncbi.nlm.nih. gov/sra/SRP083777 | Publicly available at the NCBI Short Read Archive (accession no: SRP083777) |

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
