## [Decision Letter]

Thank you for submitting your article "Transcriptional rewiring over evolutionary timescales changes quantitative and qualitative properties of gene expression" for consideration by *eLife*. Your article has been reviewed by two peer reviewers, and the evaluation has been overseen by Naama Barkai as the Senior Editor and Reviewing Editor. The following individuals involved in review of your submission have agreed to reveal their identity: Judith Berman (Reviewer #2).

The reviewers have discussed the reviews with one another and the Reviewing Editor has drafted this decision to help you prepare a revised submission.

As you will see below, the reviewers found your results highly interesting. The question of how transcription networks are rewired in evolution is fundamental, and your comprehensive analysis of the GAL system in *Candida* gives fascinating new insights into this model system. The reviewers therefore agreed that the manuscript provides an important contribution. Before publication, however, please revise the paper to account for the main comments:

1) Revisit the Cph1 findings more directly to explain previous findings.

2) Report growth phenotype of the *Y. lipolytica* rtg1 mutant.

3) Test whether GAL1/7/10 facilitate growth on GlcNAc.

4) Discuss possible implications of your study to *Candida* biology.

Reviewer #1:

Understanding how transcriptional networks evolve over time is a fundamentally important problem in biology. The authors provide a compelling and detailed example of how the GALactose utilization network of the pathogenic yeast *Candida albicans* differs from the well-characterized eukaryotic model network in the baker's yeast *S. cerevisiae*. Although differences between the regulation of these two networks were first characterized in 2007 by another lab (PMID: 17540568), it appears that the original paper may have only correctly identified half of the events. Specifically, the original study and the present study make it clear that the transcriptional activator Gal4 (and here also the co-repressor Gal80) do not regulate the GAL genes in *C. albicans*. In addition to showing that Gal4 regulated non-GAL targets in *C. albicans*, the original study identified Cph1 (the ortholog of Ste12) as the transcriptional activator of the GAL genes. In contrast, the present study identified Rtg1 and Rtg3 as GAL regulators by screening a library of transcription factor knockouts for genes whose removal eliminated galactose catabolism (in the presence of the respiration inhibitor Antimycin A). Both rtg1 and rtg3 deletions had strong phenotypic effects in this condition, while cph1 and gal4 deletions did not. Induction of a GAL1 reporter construct and a synthetic promoter containing Rtg1 and Rtg3 binding sites were shown to depend on functional RTG1 and/or RTG3. The present study also provides a much more thorough, precise, and quantitative characterization of the induction parameters of the *C. albicans* network than previous semi-quantitative studies published in 2007-2010 (which are properly cited here). Finally, by creating a rtg1 knockout in the early-diverging *Saccharomycotina Yarrowia lipolytica*, they provide evidence that Rtg1-mediated regulation of the GAL network is ancestral (although the quantitative effect of deletion is only 3-fold).

In general, this is study is a rigorous and exceptionally interesting contribution to our understanding of the evolution of transcriptional networks. Its reach will be extended because of the iconic status of the GAL network in molecular biology and because of previous evolutionary studies on this network. I found the inference (based on reporter gene expression in knockouts) that Gal1 may still play a signaling role that does not involve Gal4 and Gal80 particularly surprising. It may have interesting implications for how the downstream regulators switched, nicely setting the stage for future studies. Similarly, the result that GlcNAc may regulate galactose metabolism is thought provoking and will likely to be extended in future work. For this study, the most pertinent issues that should still be addressed are:

1) I think they owe it to the authors of the original study and the community to revisit some of the Cph1 findings more directly. The original study did not test growth of the cph1 knockout (as done here), but they did test its impact on a GAL10 reporter gene (here GAL1 was the reporter). The original study also deleted the putative Cph1 binding site and showed an effect. Did they actually unknowingly delete a Rtg1/Rtg3 binding site? Is it possible that Cph1 plays a quantitative but non-essential role in regulating GAL gene expression?

2) GEO and SRA accession numbers need to be provided.

3) The documentation of the statistical analyses and number of replicates is inadequate for a journal that prides itself on transparent reporting in this area. Generally speaking, statistical tests were reported, and the number of replicates appears "at least" two or three. In many cases, this may be because the authors feel the result is clear enough to trust, but in others (e.g. Figure 3 results with rtg3, see below), that is far from obvious. Even if there are limited numbers of replicates on the same day, there are ways to combine results across experiments done on different days. Non-parametric tests, such as Wilcoxon rank sum tests, may be particularly well suited since they do not make assumptions that are often violated in molecular biology studies.

Reviewer #1 (Additional data files and statistical comments):

See major comment #3 above.

Reviewer #2:

This is a very nice manuscript from the Johnson group that addresses the rewiring of the GAL1/7/10 genes in *C. albicans*. This issue has been addressed in prior studies almost a decade ago, but this paper extends the prior work with a comprehensive analysis of the transcription factors involved. Here, the *C. albicans* GAL1/7/10 genes are shown to be necessary for growth on galactose and to be important for formation of robust biofilms in a rat catheter model. They then identify two transcription factors Rtg1 and Rtg3 as the only two TFs that are required for growth on Galactose. They also show that the GAL promoter region and the RTG binding sites are required for the majority of GAL gene induction using GFP reporter assays.

An elegant competition experiment is performed with a high throughput flow cytometry assay that can measure GAL-GFP expression levels in *S. cerevisiae* and *C. albicans* with the *C. albicans* cells distinguished by an mCherry reporter. They detect the classic bimodal population of *S. cerevisiae* cells (in low Galactose with little or no glucose). In contrast, *C. albicans* expresses GAL genes earlier, and in a continuous manner. In addition, GAL gene transcription induction is higher as determined by RNA seq analysis. Two genes other than GAL1,7,10 are induced in both *C. albicans* and *S. cerevisiae* but we are not told which genes these are and it was not readily extracted from the supplementary tables.

They then show that the GAL1 control region is induced by GlcNAc and that the two Rtg1/3 binding sites are sufficient to drive most or all of this induction. GlcNAc was previously shown to activate GAL genes (Gunasekera et al), but this study adds the Rtg binding sites to the story. Nonetheless, it begs an important, unanswered question: Do GAL1/7/10 facilitate growth on GlcNAc? It seems that this experiment (growing Gal∆ strains on GlcNAc) would be simple to do and that it would allow consolidation of Figure 3 and Figure 6.

What is the source of galactose that cells must sense and metabolize or do these GAL genes also metabolize something else?

Along these lines, it would be useful to discuss why galactose metabolism is critical for growth as a biofilm? Is there a lot of GlcNAc in biofilms?

It also would be interesting to know if the three *C. albicans* GAL genes can function in *S. cerevisiae*. (This would be quite straightforward to do, as there are no CUG codons in any of the three genes.)

Finally, the authors connect the rewiring theme to *Yarowia lipolytica –* a very distant relative of *C. albicans* and *S. cerevisiae* and show that the only Rtg1-like gene in this organism is required for expression of GAL1. Here again, it would be interesting to know if the *Y. lipolytica* rtg1 mutant is also defective for growth on Galactose and on GlcNAc or other carbon sources.

Prior studies suggested the involvement of Cph1 or Rgt1 while these factors did not come out of the deletion library screens here. Why do the authors think that this is the case? Were very different alleles or assays used? Did the Cph1 and Rgt1 fall into the 104 colonies bin in Figure 3? Given that the experiments clearly show that Rtg1/Rtg3 are major, but not exclusive, regulators of the GAL1/7/10 genes, it would be useful to discuss the possible other mechanisms that regulate expression of these genes.

Reviewer #2 (Additional data files and statistical comments):

I think providing information about the other two genes that overlap between *S. cerevisiae* and *C. albicans* is essential. Determining if gal mutant cells have growth defects, other than not growing on galactose medium, would also provide biological context for the evolutionary results. Given that this rewiring of gal genes in general has been discussed in several other papers, this paper should extend the implications of the rewiring to the biology of the organisms.

---

## [Author Response]

As you will see below, the reviewers found your results highly interesting. The question of how transcription networks are rewired in evolution is fundamental, and your comprehensive analysis of the GAL system in Candida gives fascinating new insights into this model system. The reviewers therefore agreed that the manuscript provides an important contribution. Before publication, however, please revise the paper to account for the main comments:

1) Revisit the Cph1 findings more directly to explain previous findings.

Because both reviewers raised this point, we would like to comment on it more fully here. First we note that Martchenko et al. (Current Biology, 2007) state that their data “suggests” a role of CPH1 in regulating the GAL genes in *C. albicans*; this is not a hard conclusion. The suggestion is based on two experiments: (1) a “CPH1-like” cis-regulatory sequence was identified in the GAL promoters; when deleted in the GAL10 promoter, the expression of a GAL10 reporter was reduced. Our data agree with this; the Rtg1-3 site we identified overlaps with the “CPH1-like” cis regulatory sequence identified by Martchenko et al. Hence, the Martchenko et al. deletion result is fully consistent with Rtg1-Rtg3 being the primary regulators of the GAL genes. (2) In a CPH1 deletion strain, expression of the GAL10 reporter gene was reduced. We note that this could be an indirect effect–no direct binding of CPH1 to the GAL genes was reported. In our hands, a freshly-made CPH1 deletion strain, in comparison with a genetically matched parental strain, showed no reduction in GAL1 expression (data not shown) and no growth phenotype on media that required galactose metabolism (Figure 3, [Supplementary-material SD1-data]).

Although we cannot rule out a contribution of CPH1 to GAL10 expression, we believe that all evidence points strongly to Rtg1-Rtg3 being the primary regulators of the GAL genes in *C. albicans*. The fact that the Rtg1-3 cis-regulatory site resembles a CPH1 site accounts, we believe, for the provisional implication of CPH1 in regulation of the GAL genes.

These points have been addressed, in shorter form, in the text in the subsection “Do Rtg1 and Rtg3 directly regulate the GAL genes in C. albicans?”

2) Report growth phenotype of the Y. lipolytica rtg1 mutant.

There is no growth phenotype of the rtg1 mutant in *Y. lipolytica*; this has now been included as Figure 7—figure supplement 1. This result is discussed in the second paragraph of the subsection “Inferring the evolutionary transition of GAL regulation”.

3) Test whether GAL1/7/10 facilitate growth on GlcNAc.

We see no growth effects of Gal1/7/10 deletions on GlcNAc as the sole sugar source; these data have been included as Figure 6—figure supplement 1 and are discussed in the first paragraph of the subsection “Comparing the signals used to induce GAL genes in *C. albicans* to those used to induce the GAL genes in *S. cerevisiae*”.

4) Discuss possible implications of your study to Candida biology.

Unfortunately, there is not a lot of solid information regarding the roles and preponderance of galactose in the mammalian hosts of *C. albicans*. We are reluctant to speculate too much, as these quickly become “just-so” stories, but we have added a few lines to the Discussion addressing the possible implications to *C. albicans* biology (fifth and sixth paragraphs).

Reviewer #1:

In general, this is study is a rigorous and exceptionally interesting contribution to our understanding of the evolution of transcriptional networks. Its reach will be extended because of the iconic status of the GAL network in molecular biology and because of previous evolutionary studies on this network. I found the inference (based on reporter gene expression in knockouts) that Gal1 may still play a signaling role that does not involve Gal4 and Gal80 particularly surprising. It may have interesting implications for how the downstream regulators switched, nicely setting the stage for future studies. Similarly, the result that GlcNAc may regulate galactose metabolism is thought provoking and will likely to be extended in future work. For this study, the most pertinent issues that should still be addressed are:

1) I think they owe it to the authors of the original study and the community to revisit some of the Cph1 findings more directly. The original study did not test growth of the cph1 knockout (as done here), but they did test its impact on a GAL10 reporter gene (here GAL1 was the reporter). The original study also deleted the putative Cph1 binding site and showed an effect. Did they actually unknowingly delete a Rtg1/Rtg3 binding site? Is it possible that Cph1 plays a quantitative but non-essential role in regulating GAL gene expression?

As described above, this reviewer, we believe, is correct about the site deletion.

2) GEO and SRA accession numbers need to be provided.

SRA accession numbers, SRP083773 and SPR083777, have now been provided in the Methods, subsection “RNA-Sequencing and Analysis”.

3) The documentation of the statistical analyses and number of replicates is inadequate for a journal that prides itself on transparent reporting in this area. Generally speaking, statistical tests were reported, and the number of replicates appears "at least" two or three. In many cases, this may be because the authors feel the result is clear enough to trust, but in others (e.g. Figure 3 results with rtg3, see below), that is far from obvious. Even if there are limited numbers of replicates on the same day, there are ways to combine results across experiments done on different days. Non-parametric tests, such as Wilcoxon rank sum tests, may be particularly well suited since they do not make assumptions that are often violated in molecular biology studies.

We have included more information explaining our statistical analysis in the Methods (subsection “Flow Cytometry”). We compared the distributions in row 2 and row 4 of Figure 3 using the Wilcoxon rank sum tests (as suggested by the reviewer). From this analysis, there was insufficient evidence to reject the hypothesis that the two samples are equivalent. As such, we have changed the main text and figure to state that the differences in expression of GAL1 between a RTG3 knockout strain and its parental strain are not statistically significant (subsection “Identification of transcriptional regulator(s) controlling galactose metabolism in *C. albicans*”, second paragraph). We note that this observation does not change any of our major conclusions, particularly since ChIP experiments in *C. albicans* directly show that Rtg3 is bound to the GAL genes.

Reviewer #1 (Additional data files and statistical comments):

See major comment #3 above.

We now state the number of biological replicates for each experiment and we explicitly state how the error bars were determined (subsection “Flow Cytometry”). We did perform sufficient replicates to carry out a number of non-parametric tests. As described above, the difference between the Rtg3 and its parental strain is not statistically significant using the Wilcoxon rank sum test.

Reviewer #2:

This is a very nice manuscript from the Johnson group that addresses the rewiring of the GAL1/7/10 genes in C. albicans. This issue has been addressed in prior studies almost a decade ago, but this paper extends the prior work with a comprehensive analysis of the transcription factors involved. Here, the C. albicans GAL1/7/10 genes are shown to be necessary for growth on galactose and to be important for formation of robust biofilms in a rat catheter model. They then identify two transcription factors Rtg1 and Rtg3 as the only two TFs that are required for growth on Galactose. They also show that the GAL promoter region and the RTG binding sites are required for the majority of GAL gene induction using GFP reporter assays.

An elegant competition experiment is performed with a high throughput flow cytometry assay that can measure GAL-GFP expression levels in S. cerevisiae and C. albicans with the C. albicans cells distinguished by an mCherry reporter. They detect the classic bimodal population of S. cerevisiae cells (in low Galactose with little or no glucose). In contrast, C. albicans expresses GAL genes earlier, and in a continuous manner. In addition, GAL gene transcription induction is higher as determined by RNA seq analysis. Two genes other than GAL1,7,10 are induced in both C. albicans and S. cerevisiae but we are not told which genes these are and it was not readily extracted from the supplementary tables.

The genes have been named in the last paragraph of the subsection “Comparing the genes induced by galactose in *C. albicans* to those induced in *S. cerevisiae*” and are also annotated with asterisks in [Supplementary-material SD8-data]. The text now reads:

“Except for the GAL1, GAL7 and GAL10 genes, there was little overlap (only 2 additional genes, RHR2 and GPP1, annotated in *S. cerevisiae* as enzymes involved in sugar metabolism) between the *S. cerevisiae* and *C. albicans* galactose-induced genes (Figure 5—figure supplement 2, [Supplementary-material SD8-data]).”

They then show that the GAL1 control region is induced by GlcNAc and that the two Rtg1/3 binding sites are sufficient to drive most or all of this induction. GlcNAc was previously shown to activate GAL genes (Gunasekera et al), but this study adds the Rtg binding sites to the story. Nonetheless, it begs an important, unanswered question: Do GAL1/7/10 facilitate growth on GlcNAc? It seems that this experiment (growing Gal∆ strains on GlcNAc) would be simple to do and that it would allow consolidation of Figure 3 and Figure 6.

We have included this experiment as Figure 6—figure supplement 1. Here we took independently constructed knockouts strains and a matched wild-type and spotted them onto media containing 2% GlcNAc. We did not observe a growth defect of the mutants compared to wild-type strains on GlcNAc. This is discussed in the first paragraph of the subsection “Comparing the signals used to induce *GAL* genes in *C. albicans* to those used to induce the *GAL* genes in *S. cerevisiae*” and in the Methods, subsection “Plating Assays”.

What is the source of galactose that cells must sense and metabolize or do these GAL genes also metabolize something else?

This is a very good question, but the answer is not known in any detail. Galactose is synthesized in the human body and used as components of glycolipids and glycoproteins. Moreover, galactose can be ingested through diet, most notably through the intake of lactose (Lactose is hydrolyzed into glucose and galactose).

Given that *C. albicans* inhabits many niches in its warm-blooded hosts and given that the GAL genes have been maintained by selection, it is highly likely that *C. albicans* responds to galactose in its natural environment.

As far as we know, there is no evidence that the GAL genes metabolize something else, but there is also no direct evidence against this idea.

We have discussed these issues in the Discussion (fifth and sixth paragraphs).

Along these lines, it would be useful to discuss why galactose metabolism is critical for growth as a biofilm? Is there a lot of GlcNAc in biofilms?

This is an intriguing question. Galactose is not one of the major carbohydrates found in the extracellular matrix of biofilms (glucose and mannan, respectively, see Mitchell and Andes, PNAS, 2015). Moreover, it doesn’t appear that GlcNAc, the fundamental repeating unit of chitin, found in many cell walls, is thought to be part of the extracellular matrix of *C. albicans* biofilms (see Mitchell et al., Adv Exp Med Biol, 2016). It is, of course, possible that even though galactose and GlcNAc are not thought to be major components of the *C. albicans* extracellular matrix, *C. albicans* cells still sense and respond to galactose and or GlcNAc elsewhere in the biofilm but there is limited information regarding this available. Hence, we have chosen to not include this in the text.

It also would be interesting to know if the three C. albicans GAL genes can function in S. cerevisiae. (This would be quite straightforward to do, as there are no CUG codons in any of the three genes.)

If the *C. albicans* genes could function in *S. cerevisiae*, it would confirm the conclusion, based on many lines of evidence, that the GAL genes have maintained the same function. However, if they didn’t work, there would be many possible trivial explanations. We feel that, given the conservation of the GAL genes across a very wide phylogenic distribution, and the fact that diverse species require these genes to grow on galactose is very strong evidence they can carry out the same function.

Finally, the authors connect the rewiring theme to Yarowia lipolytica – a very distant relative of C. albicans and S. cerevisiae and show that the only Rtg1-like gene in this organism is required for expression of GAL1. Here again, it would be interesting to know if the Y. lipolytica rtg1 mutant is also defective for growth on Galactose and on GlcNAc or other carbon sources.

We have included this experiment as Figure 7—figure supplement 1 and in second paragraph of the subsection “Inferring the evolutionary transition of GAL regulation”. Here we took knockouts strains and a matched wild-type and spotted them onto media containing 2% galactose. We did not observe a strong growth defect compared to wild-type strains on galactose. We note that *Y. lipolytica* is an obligate respirator; meaning that the use of Antimycin A (or any respiration inhibitor) for a growth assay (similar to what was done in *S. cerevisiae* and *C. albicans*) is not possible. This is noted in the Methods in the subsection “Plating assays”.

Prior studies suggested the involvement of Cph1 or Rgt1 while these factors did not come out of the deletion library screens here. Why do the authors think that this is the case? Were very different alleles or assays used? Did the Cph1 and Rgt1 fall into the 104 colonies bin in Figure 3? Given that the experiments clearly show that Rtg1/Rtg3 are major, but not exclusive, regulators of the GAL1/7/10 genes, it would be useful to discuss the possible other mechanisms that regulate expression of these genes.

Cph1 and Rgt1 were previously implicated as important for the regulation of GAL gene expression using motif analysis. As described above, the “CPH1-like’ motif overlaps extensively with the recognition site for RTG1-RTG3. In previous microarray studies, it was found that *C. albicans* genes induced by galactose are enriched for Rgt1 motifs in their promoters; however, no follow-up studies, to our knowledge, have been important. Neither Cph1 or Rgt1 are required for growth on galactose or for full expression of GAL1 (data not shown); they were included amongst the 104 deletion strains with no effect in Figure 3 (they are both in the bin to the very right), the complete data being available in [Supplementary-material SD1-data]). It is possible, as pointed out by the reviewer, that Rgt1 makes a contribution to GAL1 expression that was not apparent in our experiments, perhaps in a special type of environment. However, RTG1 and RTG3 have much larger effects when growth on galactose is monitored.

These points have been addressed, in shorter form, in the last paragraph of the subsection “Do Rtg1 and Rtg3 directly regulate the *GAL* genes in *C. albicans*?”.